# Understanding the Effects of Neuron Dominance in Deep Reinforcement Learning

**Zifan Wu**        *zifan.w@utah.edu*
*University of Utah*

**Qian Lin**
*University of Utah*

**Blake Lawlor**
*University of Utah*

**Haijun Zhao**
*Sun Yat-sen University*

**Daniel S. Brown**
*University of Utah*

**Reviewed on OpenReview:** *https://openreview.net/forum?id=VNV1h77UnH*

## Abstract

Recent studies in deep reinforcement learning have revealed that neural networks tend to lose their capacity to adapt to new targets over the course of training. The proliferation of inactive neurons, i.e., the so-called "dormant neurons", has been identified as one source of capacity loss. This paper investigates *dominant neurons*, neurons whose activation values are significantly larger than average, as a potential cause for neuron dormancy. We demonstrate the existence of dominant neurons in a number of visual control tasks, and perform an analysis of the learning dynamics showing how dominant neurons can induce dormancy in the subsequent layer. To gain a better understanding of this phenomenon, we examine it through the lens of representation learning and establish its connection with representation collapse. Furthermore, this paper evaluates several mitigation strategies for dominant neurons across a variety of visual control tasks. Our results show that strategies that induce lower peak activation scores tend to exhibit greater representational capacity, lower dormant neuron percentage, and better performance. Among these mitigation strategies, LayerNorm with weight decay has the strongest performance, despite its simplicity. Moreover, switching the value learning loss from regression to a classification loss also significantly mitigates the neuron dominance issue and improves the performance. As a potential explanation of the effectiveness of classification losses, we provide an analysis that shows how a classification loss can prevent representation collapse.

## 1 Introduction

Over the past decade, deep Reinforcement Learning (RL) has achieved tremendous successes in various domains (Silver et al., 2016; Andrychowicz et al., 2020; Afsar et al., 2022). However, it has been widely recognized that the performance of deep RL algorithms is still brittle and sensitive to various seemingly unimportant design choices related to neural networks (Henderson et al., 2018). Regarding this problem, a series of research endeavors have recently been undertaken to understand the interplay between the learning objective of RL and the optimization dynamics of neural networks (Kumar et al., 2020; Dohare et al., 2021; Lyle et al., 2021; 2022; Nikishin et al., 2022; Lyle et al., 2023; 2024b). As pointed out by this line of work,

a crucial issue that widely exists in deep RL methods is plasticity loss, the loss of a network's ability to overwrite its prior predictions based on newly observed experience. Research efforts have been made to reveal the mechanisms that might lead to plasticity loss; in particular, Sokar et al. (2023) demonstrate that the percentage of inactive neurons tends to increase during deep RL training, leading to declined network expressivity.

By contrast, recent work by Sun et al. (2024) reveals the existence of massive activation values in large language models, while Qin et al. (2024) identify "over-active" neurons in the multi-agent RL setting. Inspired by these works, we seek to better understand plasticity loss by studying the impact of *dominant neurons*, neurons whose activations are disproportionately larger than average. We initiate our investigation by empirically verifying the existence of large activations in visual DM-Control tasks. Our preliminary results further indicate that there is a correlation between dominant neurons and next-layer saturation in multiple sparse reward tasks. To provide a theoretical explanation for the phenomenon, we present an analysis of the process by which a dominant neuron can induce complete saturation in the subsequent layer. Delving deeper into this phenomenon, we notice its connection with previous work studying representation collapse (Kumar et al., 2020; Lyle et al., 2022), where the network outputs a zero matrix for any inputs. Our analysis shows that higher peak activation scores tend to correspond to reduced model representational capability. Empirical results further corroborate this theoretical connection by showing an inverse correlation between the magnitude of the largest activation score and the effective rank of the feature matrix, a metric used for quantifying representational capability (Kumar et al., 2020).

We then study a variety of strategies that might mitigate neuron dominance and compare their effectiveness on the visual DM-Control suite (Tassa et al., 2018). The results suggest that approaches that result in lower maximum activation scores typically exhibit greater representational capacity, lower dormant neuron percentage, and better performance, which offers a new perspective to understand the effectiveness of these approaches. Among these mitigation strategies, LayerNorm with weight decay obtains the strongest performance, despite its simplicity. By definition, LayerNorm shifts the preactivation distribution towards zero mean and unit variance. Our empirical investigation shows that this strategy can eliminate dominant neurons and revive a completely saturated layer (universally negative preactivations). Moreover, we find that switching from the traditional regression loss to, HL-Gauss, a classification loss (Farebrother et al., 2024) in value learning also performs well especially in sparse reward tasks. Inspired by this finding, we provide an analysis that shows how a classification loss can prevent representation collapse. Given that HL-Gauss has been demonstrated to yield significant performance improvement across various RL benchmarks and network architectures, our findings provide a step towards better understanding the benefits of using this loss, rather than the traditional regression loss, in deep RL. Taken together, our results suggest that neuron dominance may provide a novel mechanistic explanation for several previously observed failure modes and fixes in deep RL, including representation collapse, the benefits of normalization, and the effectiveness of classification-based value learning.

The contributions of this work can be summarized as follows:

1. We identify dominant neurons as a previously overlooked mechanism that contributes to plasticity loss in deep RL;

2. We establish a theoretical connection between neuron dominance and diminished network representational capacity. We also show empirical evidence for this theory by artificially inducing neuron dominance and showing that it leads to next-layer saturation;

3. We analyze several strategies that might mitigate neuron dominance, and empirically evaluate them across a variety of tasks. The results show that while some strategies excel in non-sparse reward tasks, others are better in sparse-reward tasks, and that overall HL-Gauss and LayerNorm with Weight Decay perform the best;

4. We provide theoretical proof that classification losses exhibit inherent advantages against representation collapse.

## 2   Background

**MDPs & RL.**   A Markov decision process (MDP) $\mathcal{M}$ is defined by tuple $(\mathcal{S}, \mathcal{A}, R, P, \rho_0, \gamma)$, where $\mathcal{S}$ is the state space, $\mathcal{A}$ is the action space, $R(s, a)$ is a scalar reward function of $s \in \mathcal{S}$ and $a \in \mathcal{A}$, $P(s'|s, a) \in [0, 1]$ is the transition dynamics which is assumed to be unknown, $\gamma \in [0, 1)$ is the discount factor and $\rho_0$ is the initial state distribution. A policy $\pi(a|s)$ maps states to a probability distribution over the action space. The expected value of choosing action $a$ under state $s$ and following policy $\pi$ afterwards is defined by $Q^\pi(s, a) = \mathbb{E}_{\pi, P}\left[\sum_t \gamma^t R(s_t, a_t)\right]$. Typically, RL methods learn a value function using temporal difference learning (Sutton, 1988): $\arg\min_{\hat{Q}} \mathbb{E}\left[\left(\hat{Q}(s, a) - (R(s, a) + \gamma \hat{Q}(s', a'))\right)^2\right]$, where $s', a'$ are the next state and next action according to the transition dynamics and the current policy. In general, the goal of RL is to learn the optimal policy $\pi^*$ which maximizes the expected discounted sum of rewards from the environment.

**Dormant/Saturated/Zombie Neurons.**   In deep RL, the value function is parameterized by a neural network $Q_\theta$. Figure 1 shows an example diagram, and our corresponding notation, for two layers in a neural network. We denote $N_k$ as the width of layer $k$, and $g_i^k(x)$ as the activation of the $i$-th neuron of layer $k$ under input $x \sim D$ where $D$ is the input distribution. We denote the weight connecting the $i$-th neuron in layer $k$ to the $j$-th neuron in layer $k+1$ as $w_{ij}^{k+1}$. According to the definition in Sokar et al. (2023), a neuron is labeled *dormant* when its *activation score*, defined as

$$u_i^k = \frac{\mathbb{E}_{x\sim D}[g_i^k(x)]}{\frac{1}{N_k}\sum_{j=1}^{N_k} \mathbb{E}_{x\sim D}[g_j^k(x)]}, \tag{1}$$

Figure 1:   Notation for network activations and weights.

is lower than or equal to a pre-set threshold (we use 0.1 in our experiments). The scores are normalized such that they sum to 1 within a layer. This makes the comparison of neurons in different layers possible. Given an input distribution, we refer to neurons whose preactivations are always non-positive as *saturated neurons*, and those whose preactivations are always positive as *zombie neurons*, as termed by Lyle et al. (2024b).

According to the chain rule, saturated neurons induce a vanishing gradient pathology, effectively decoupling the objective function from the parameter updates in early layers and leading to a precipitous decline in the network's functional expressivity. In contrast, becoming a zombie neuron signifies the "linearization" of the neuron, a state where a unit functions as a linear (or quasi-linear) operator on its inputs. This occurs, for instance, when a ReLU gate is restricted to positive inputs—rendering it an identity mapping—or when smooth activations encounter low-variance inputs, causing them to approximate linear functions. These "zombie units" differ from saturated units in that they facilitate, rather than obstruct, gradient flow and signal propagation. However, an abundance of these units inevitably diminishes the model's effective representational capacity.

## 3   Dominant neurons and their impact

In prior work by Sokar et al. (2023), the proliferation of dormant neurons is identified as a mechanism by which deep RL networks lose their expressivity. Dormant neurons (Sokar et al., 2023) are defined as neurons that are significantly less active than other neurons in the same layer. In contrast, Sun et al. (2024) study neurons that have extremely high activation values in LLMs, and Qin et al. (2024) define "over-active" neurons as neurons that have expected activation scores larger than a fixed pre-set value, in the multi-agent RL setting. Inspired by these recent works, we start with an empirical investigation into the existence and effect of large activations in standard single-agent RL. While dormant neurons describe a loss of activity, *dominant* neurons capture a complementary failure mode in which excessive activity in a small subset of neurons suppresses learning dynamics elsewhere in the network. Our preliminary results show that excessively large activations are usually accompanied by the complete saturation of the subsequent layer. We then analyze the learning dynamics in the presence of dominant neurons, thereby explaining the phenomena

observed in our experiments. To better understand this phenomenon, we further investigate its theoretical connection to representation collapse, given the similarities between these two phenomena.

### 3.1 Phenomenon: Excessively large activations co-occur with complete saturation of neurons in the subsequent layer

In contrast to the dormant neurons studied in (Sokar et al., 2023), we investigate the opposite scenario: what challenges emerge when certain neurons demonstrate significantly higher activation levels compared to other neurons in the same layer? To answer this question, we conduct empirical experiments in the visual DM-Control suite (Tassa et al., 2018), where agents must learn from high-dimensional pixel inputs, to identify neurons exhibiting anomalously large activations across network layers. We adopt DrQ (Yarats et al., 2021), a widely used strong baseline RL algorithm, as our backbone algorithm. Building upon SAC Haarnoja et al. (2018), DrQ uses image augmentation techniques to deal with pixel inputs and obtains significant improvement over SAC.

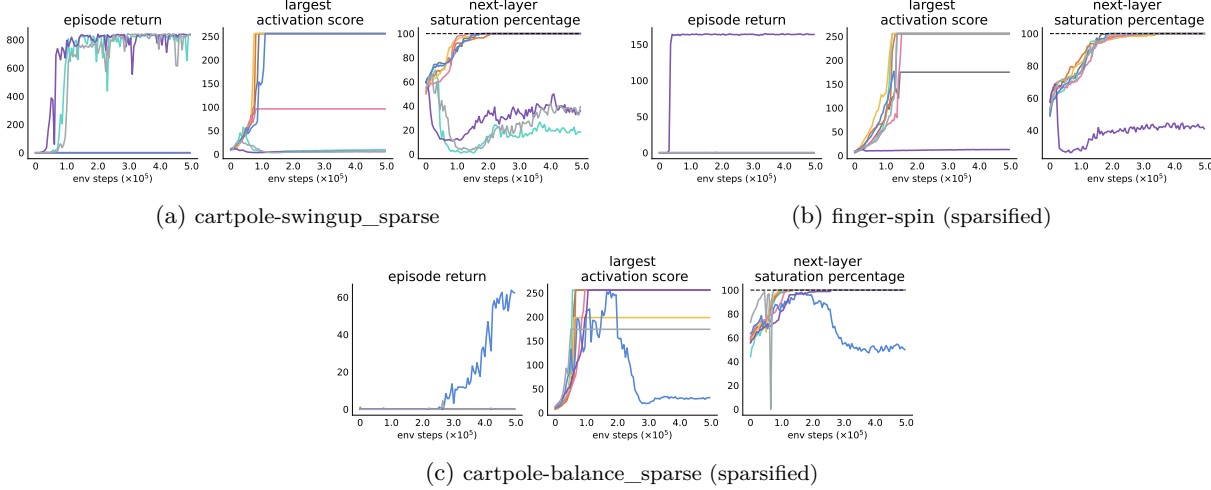

Figure 2: Results of 8 random seeds in 6 sparse reward tasks showing strong correlation between the largest activation score and the next-layer saturation percentage. Note that for demonstration purposes, here we show the statistics of 8 single runs separately.

In our experiments, DrQ performs well in many tasks, but often fails to learn good policies in sparse-reward tasks, converging to a policy that obtains zero return. As we look into the activation statistics of the value network in these sparse-reward tasks, we observe neurons with extremely large activation scores in the layer before the penultimate layer [1]. As shown in Figure 2 and 3, there is a drastic increase in the largest activation score, followed by a complete saturation of its next layer, i.e., the penultimate layer. In addition to the standard DrQ architecture, we also observe this phenomenon when replacing the CNN layers with ResNet-18 (He et al., 2016) (see Appendix C.1). This signifies a strong correlation between large activation scores and full-layer saturation, which motivates our analysis of the learning dynamics in the next subsection.

### 3.2 Impact of a dominant neuron on its subsequent layer: An analysis of the learning dynamics

In this section, we formally define neuron dominance and analyze its effect on neuron activations in subsequent layers. For our analysis of neural network value learning, we consider networks consisting of at least two hidden fully-connected layers using ReLU (Agarap, 2018) as the activation function, which is also a common setting in RL (Haarnoja et al., 2018). We let layer $k + 1$ be the last hidden layer (followed by a linear final layer). We also assume that the network is trained for a regression task using the MSE loss, which is the standard learning task for RL value networks (Sutton & Barto, 1998).

---

[1]The phenomenon is originally observed in Cartpole-swingup_sparse. To better study this, we modify the other three tasks to create more sparse-reward tasks. See Appendix B for more details related to the environments.

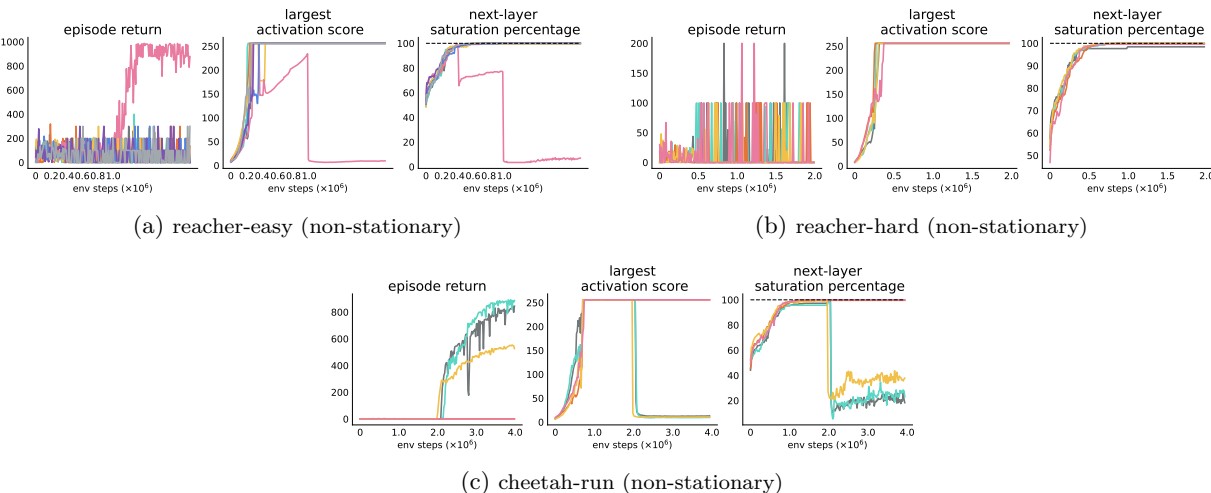

(a) reacher-easy (non-stationary)  (b) reacher-hard (non-stationary)

(c) cheetah-run (non-stationary)

Figure 3: Results of 8 random seeds in 6 sparse reward tasks showing strong correlation between the largest activation score and the next-layer saturation percentage. In these three tasks, the reward is masked to 0 during the first half of training, and is unmasked in the second half.

**Definition 1.** *Given an input distribution $D$, a neuron $i$ in layer $k$ is defined as a **dominant neuron** if its activation score satisfies: $u_i^k > \frac{N_k}{2}$.*

This definition captures the regime in which a single neuron's expected activation dominates the aggregate contribution of all others, enabling clean theoretical analysis of its downstream effects. The condition $u_i^k > \frac{N_k}{2}$ immediately implies $\mathbb{E}[g_i^k(x)] > \sum_{j \neq i} \mathbb{E}[g_j^k(x)]$, i.e., in expectation, the sum of all the other activations in the same layer is less than the dominant activation alone. The threshold for being a dominant neuron, i.e., half of the layer width, results from our later analysis, where it guarantees dominance in the activation values of the next layer. In practice, we observe such dominant neurons in several sparse-reward tasks, as shown in Figure 2 and 3. Since the width for layer $k$ is 256 in our experiment, most largest activation scores in Figure 2 and 3 satisfy the condition in Definition 1.

In the following analysis, without loss of generality, we denote the first neuron as the dominant neuron (i.e., $i = 1$). Our first result regarding dominant neurons is Theorem 1, which shows that, under certain conditions, there would only be saturated or zombie neurons in the layer following the layer that contains a dominant neuron. This provides a theoretical basis for the phenomenon observed in Section 3.1. Note that all full proofs in this paper can be found in Appendix A.

**Theorem 1.** *Assume that for any neuron $m$ in layer $k+1$, the weight $w_{1m}^{k+1}$ connecting a dominant neuron in layer $k$ and this neuron is no less than the average of weights from the same layer that connect to this neuron, i.e., $w_{1m}^{k+1} \geq \frac{1}{N_k} \sum_{i=1}^{N_k} w_{im}^{k+1}$, and that the biases in layer $k+1$ are zero. Under these assumptions, a neuron in layer $k+1$ will either be a saturated or a zombie neuron.*

The intuition here is that the sign of a preactivation in layer $k+1$ depends on the sign of the weight that connects this neuron with the dominant neuron. Further, we show that *given a sufficiently large learning rate $\left(\frac{w_{1j}^{k+1}}{g_1^k(x)(\hat{y}-y)w_j^{k+2}}\right.$, notations explained below), all the outgoing weights of the dominant neuron eventually turn negative, and thus layer $k+1$ becomes fully saturated*: According to the chain rule, a positive outgoing weight of the dominant neuron, say $w_{1j}^{k+1}$, receives gradient updates as follows:

$$w_{1j}^{k+1} \leftarrow w_{1j}^{k+1} - \alpha(\hat{y} - y)w_j^{k+2}g_1^k(x), \qquad (2)$$

where $w_j^{k+2}$ connects $g_j^{k+1}$ to the output node $\hat{y}$, and $y$ is the target given input $x$. The basic idea is that: 1) when $\hat{y} - y > 0$ and $w_j^{k+2} < 0$, $w_j^{k+2}$ will be updated towards a smaller value to decrease $\hat{y}$, so eventually

Figure 4: Results of introducing a dominant neuron during supervised training on CIFAR10. The dominant neuron is created when the validation accuracy reaches 0.5. The results show that as the largest activation score rapidly rises, the subsequent layer soon becomes fully saturated, and the validation accuracy drops dramatically. These results match the prediction in Theorem 1.

there will be a moment when $(\hat{y} - y)$ and $w_j^{k+2}$ have the same sign, and vice versa; 2) when $(\hat{y} - y)$ and $w_j^{k+2}$ have the same sign and $\alpha \geq \frac{w_{1j}^{k+1}}{g_1^k(x)(\hat{y}-y)w_j^{k+2}}$, $w_{1j}^{k+1}$ will become negative according to Equation 2.

Together, the analysis above provides a theoretical explanation of the phenomenon shown in Section 3.1. However, given the definition of the dominant neuron, the condition becomes harder to satisfy as the layer width increases. In the next subsection, we will generalize this analysis by studying how the activation score generally relates to the representational capacity of the network.

We directly simulate the effect predicted by Theorem 1 by artificially constructing a dominant neuron during training. Since the activation score is partly determined by the data distribution which keeps changing during online RL, it is infeasible to manually construct a dominant neuron during this process. As an alternative, we conduct an intervention experiment on CIFAR10 where the data distribution is fixed. The dominant neuron is created by scaling up the incoming weights of the neuron that has the largest activation score (see Appendix B.3 for the experimental details). Figure 4 verifies the prediction in Theorem 1 by showing neuron saturation of the subsequent layer after the dominant neuron is created.

**Remark:** The assumptions in Theorem 1 are sufficient conditions for the theoretical result. In all of our experiments on DM-Control, we use the standard DrQ network (a shared 4-layer CNN encoder, two 3-layer MLP heads for actor and critic, repectively) that do include biases. The phenomenon expected by the result of Theorem 1 is still observed in practice without fully satisfying the conditions.

### 3.3 Connection with representational capacity

It has been previously observed that, due to reward sparsity, the network tends to output zero for all possible inputs without distinguishing them (Lyle et al., 2022). Specifically, this can be achieved by making the penultimate layer fully saturated, i.e., representation collapse (Kumar et al., 2020), which is similar to the effect of having a dominant neuron as demonstrated in previous subsections. This motivates us to leverage metrics for representational capacity to study neuron dominance in this subsection.

Prior work by Kumar et al. (2020) has reported that value networks trained in sparse reward settings are prone to *representation collapse*, a phenomenon where the bottom layers learn to use a zero feature to represent every input. They propose to use the *effective rank* to quantify representational capacity: for some threshold $\delta \in (0, 1)$,

$$\mathrm{srank}_\delta(\Phi) = \min \left\{ k : \frac{\sum_{i=1}^k \sigma_i(\Phi)}{\sum_{i=1}^N \sigma_i(\Phi)} \geq 1 - \delta \right\}, \tag{3}$$

where $\Phi \in \mathbb{R}^{|\mathcal{S}||\mathcal{A}| \times N}$ is the output of the last hidden layer across all state-action pairs, referred to as the feature matrix, and $\{\sigma_i(\Phi)\}_{i=1}^N$ are the singular values of $\Phi$ in descending order. Since in our experiments we only observe dominant neurons in the layer preceding the penultimate layer, here we adopt a broader

interpretation of the feature matrix, specifically defining it as the output of the hidden layer immediately preceding the penultimate layer. From this point of view, the dominant neuron can also be interpreted as a column of the feature matrix whose summation dominates other columns (i.e., $s_1 = \sum_i \Phi_{i1} \geq \sum_{j \neq 1} \sum_i \Phi_{ij}$ assuming, without loss of generality, that neuron 1 is the dominant neuron and where $s_1$ is the summation of the first column of the feature matrix), implying a potential connection with the concept of rank collapse.

In practice, it is usually unrealistic to compute the entire feature matrix since it involves the joint state-action space which may be enormous or even infinite. Thus, this matrix is typically approximately computed over a sampled batch of data. Regarding this sampled feature matrix, we have the following conclusion in Theorem 2 that connects neuron dominance and a reduction in the representation capacity.

**Theorem 2.** *Denote $\hat{s}_i$ as the sum of the $i$-th column of the sampled feature matrix $\hat{\Phi} \in \mathbb{R}^{B \times N}$, where $N$ is the layer width and $B$ is the batch size. Denote the singular values of the sampled matrix as $\{\hat{\sigma}_i\}$. Assume that the samples are i.i.d. and let $n = \min(N, B)$. Assuming, without loss of generality, that the index of the largest neuron is 1 (i.e., $\mathbb{E}[s_1] \geq \mathbb{E}[s_j], \forall j$), we have:*

$$\mathbb{E}\left[\frac{\hat{\sigma}_1}{\sum_{i=1}^N \hat{\sigma}_i}\right] \geq \sqrt{\frac{1}{nB} + \frac{2(B-1)}{n\left[B(1+Var(\hat{s}_i)) + \frac{\sum_{i=2}^N \mathbb{E}[\hat{s}_i^2]}{(\mathbb{E}[\hat{s}_1])^2}\right]}}. \tag{4}$$

Theorem 2 indicates that for the sampled matrix, the expected ratio of $\hat{\sigma}_1$ to the sum of all singular values is lower bounded by a quantity that increases with the escalation of the degree of neuron dominance, $\frac{\sum_{i=2}^N \mathbb{E}[\hat{s}_i^2]}{(\mathbb{E}[\hat{s}_1])^2}$, i.e., how large the largest activation is with respect to the others. Since $\delta < 1$, the smallest possible value for srank is 1. Based on the definition of srank, this is achieved when $\frac{\hat{\sigma}_1}{\sum_{i=1}^N \hat{\sigma}_i} > 1 - \delta$. Thus, we have that $Pr(\frac{\hat{\sigma}_1}{\sum_{i=1}^N \hat{\sigma}_i} > 1 - \delta)$ increases as $\mathbb{E}[\frac{\hat{\sigma}_1}{\sum_{i=1}^N \hat{\sigma}_i}]$ increases. Because increasing the degree of neuron dominance (i.e., decreasing $\frac{\sum_{i=2}^N \mathbb{E}[\hat{s}_i^2]}{(\mathbb{E}[\hat{s}_1])^2}$) increases the lower bound on $\mathbb{E}[\frac{\hat{\sigma}_1}{\sum_{i=1}^N \hat{\sigma}_i}]$, Theorem 2 provides a theoretical connection between neuron dominance and representational capacity, and also corroborates our empirical results in Section 4.1 where approaches that are capable of maintaining a low dominance degree tend to have a large srank.

Intuitively, if a dominant neuron can cause a complete representation collapse where the srank falls to 1, then as a form of relaxation, a large albeit not strictly dominant neuron may well be associated with a relatively low srank. In Section 4.1, we perform an empirical investigation on the effectiveness of different mitigation strategies for neuron dominance (see Figure 9), where strategies that induce value networks with a higher largest activation score tend to have lower sranks, providing positive evidence for this conjecture. Note that we do not claim that neuron dominance is necessary for representation collapse; rather, it provides a mechanistic pathway by which collapse can arise in value networks.

## 4 Mitigation strategies

The previous section has introduced dominant neurons, demonstrated their impact, and discussed their connection with representation collapse, a concept closely related to plasticity loss. Recently, several mitigation strategies have been shown to be effective in preserving network plasticity. Since representation collapse can be viewed as a special form of plasticity loss where the network loses all its capacity (Lyle et al., 2022), it is natural to consider if these mitigation strategies have any effect on neuron dominance which, as shown in the previous section, is a potential cause of representation collapse. In this section, we present an empirical investigation comparing the effect of these strategies on neuron dominance and the performance. The testbed for the investigation is visual DM-Control Tassa et al. (2018), which uses raw pixels as the agent's observation and contains a variety of challenging continuous control tasks. See Appendix B for implementation-level details as well as hyperparameter settings in our experiments. [2]. Below we briefly describe each mitigation strategy.

---

[2]Code available at this url.

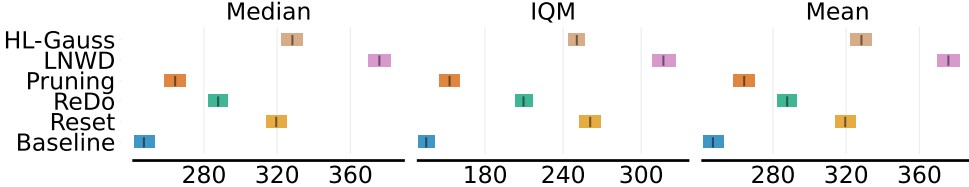

Figure 5: Median, interquartile mean (IQM), and mean with 95% bootstrapped confidence intervals of the episodic return obtained by the five tested approaches in 12 visual DM-Control tasks including 8 non-sparse reward tasks and 4 sparse reward tasks. Overall, LNWD achieves the best performance, while HL-Gauss and Reset are the second best and perform similarly.

**Reset** (Nikishin et al., 2022) periodically re-initializes the network's last few layers while preserving the experience stored in the replay buffer. By fully refreshing the layers that might contain a dominant neuron, this method seeks to, at least temporarily, recover network plasticity.

**ReDo** (Sokar et al., 2023) periodically recycles dormant neurons during training by re-initializing their incoming weights and zeroing out their outgoing weights.

**Gradual Magnitude Pruning** (Obando-Ceron et al., 2024) progressively sparsifies a dense network throughout training by pruning parameters with relatively low magnitudes.

These three approaches can be categorized as parameter-perturbation approaches since they directly modify network parameters. We also evaluate two approaches which, unlike the first three, do not directly modify parameters in the network.

**HL-Gauss** (Farebrother et al., 2024) reframes the regression problem in value function learning as classification and utilizes the cross entropy loss to optimize the parameters. It evenly divides a preset value interval into bins which represent the probabilities of taking the corresponding values. The method name comes from the label-smoothing approach it uses, called HL-Gaussian (Imani & White, 2018).

**LayerNorm with Weight Decay (LNWD)** (Lyle et al., 2024b) combines LayerNorm and Weight Decay and aims to deal with both preactivation distribution shifts and parameter norm growth.

## 4.1 Empirical evaluation

We evaluate the aforementioned mitigation approaches on the visual DM-Control suite (Tassa et al., 2018) using the DrQ implementation (for details of the architecture, see Appendix B.2). Although we have not observed neurons that satisfy Definition 1 in non-sparse reward tasks, Section 3.3 implies that even if a neuron is not strictly dominant, it could have a negative impact on the network's representational capacity. Therefore, in addition to the four sparse reward tasks tested in Section 3.1, we also evaluate these approaches on eight non-sparse reward tasks. We fine-tuned the hyperparameters of each of these approaches. The details of the hyperparameter settings are shown in Appendix B.1.

### 4.1.1 Overall performance

Figure 5 demonstrates the aggregated performance comparison between the approaches across these tasks, while the learning curves in each task are shown in Figure 6. In aggregate, LNWD tends to perform favorably among these approaches, despite its simplicity; HL-Gauss and Reset show similar aggregate performances; however, as shown in Figure 7 and Figure 8, they exhibit different strengths across task types: HL-Gauss tends to perform better in sparse-reward tasks while Reset appears stronger in non-sparse tasks. We also observe that the two approaches that directly modify the network parameters in a fine-grained manner, i.e., ReDo and Pruning, show a modest improvement over the DrQ baseline, though their gains are smaller than those achieved by the other three approaches (HL-Gauss, LNWD, Reset). We note that performance varies considerably across individual tasks (see Figure 6), and that these results are obtained using a single

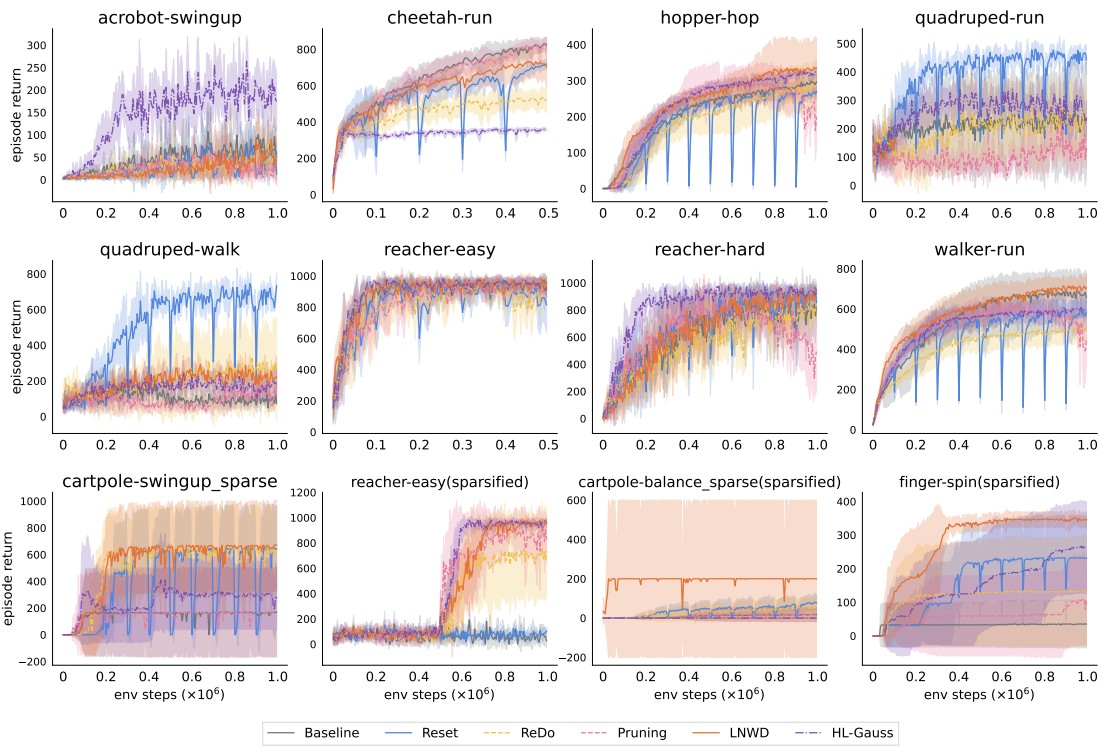

Figure 6: Learning curves on 8 non-sparse reward tasks and 4 sparse reward tasks. Curves represent the mean of runs over 6 different random seeds, and shaded regions correspond to standard deviation among these runs. LNWD performs at least decently in all tasks, while HL-Gauss and Reset excel in different sets of tasks.

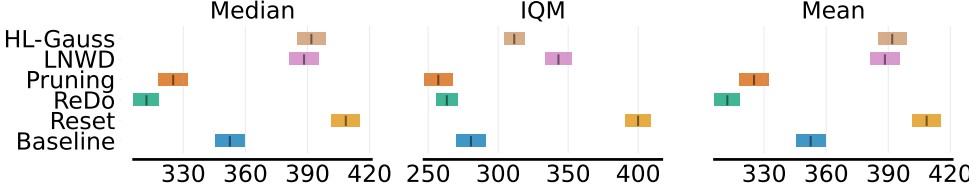

Figure 7: Median, interquartile mean (IQM), and mean with 95% bootstrapped confidence intervals of the episodic return obtained by the five tested approaches in 8 non-sparse reward tasks. Surprisingly, Reset even outperforms LNWD in this set of tasks.

algorithm (DrQ) and architecture on the DM-Control suite. Different hyperparameter configurations or algorithmic choices could plausibly alter the relative ordering of these approaches. That said, within our experimental setup, the pattern tentatively suggests that approaches with a more substantial impact on the learning dynamics may be more effective than fine-grained parameter modifications at addressing the pathological optimization dynamics studied here (Lyle et al., 2024b).

### 4.1.2 Performances in sparse and non-sparse reward tasks

Figure 7 and Figure 8 show the aggregated performance in non-sparse and sparse reward tasks, respectively. We found that in non-sparse reward tasks, Reset is the best-performing approach, surpassing LNWD; while in sparse reward tasks, the advantage of LNWD over the other approaches becomes larger. Moreover, unexpectedly, although ReDo and Pruning improve the performance in sparse reward tasks, they have a negative effect on the performance in non-sparse reward tasks. Our empirical results provide practical

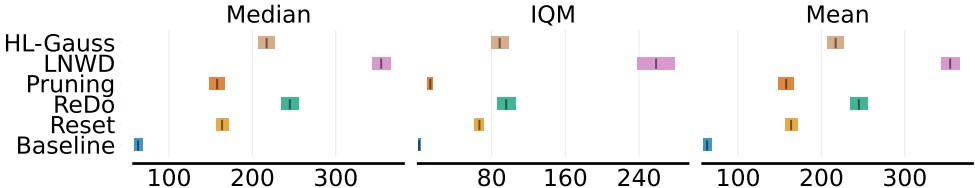

Figure 8: Median, interquartile mean (IQM), and mean with 95% bootstrapped confidence intervals of the episodic return obtained by the five tested approaches in 4 sparse reward tasks. In this set of tasks, LNWD outperforms all the other strategies by a large margin.

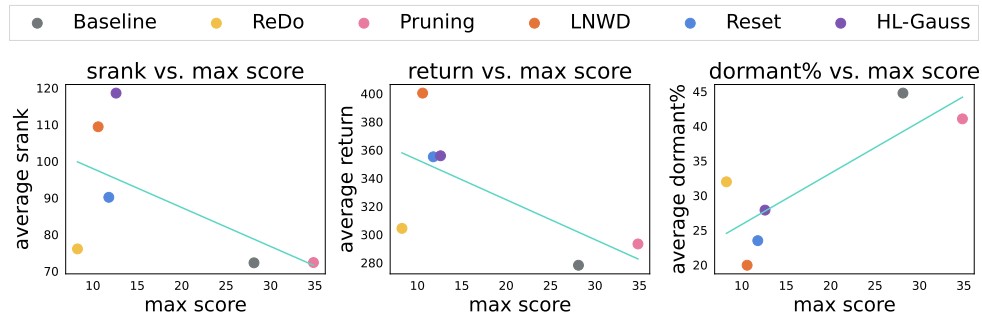

Figure 9: Results showing the correlation between the largest activation score and three metrics: srank, return, and dormant percentage, respectively. Each data point represents one approach, with values averaged across 6 random seeds and 12 tasks. Approaches that lead to a low maximum score usually have a high srank, high return, and low dormant percentage.

guidance for applying these approaches: LNWD is always a good option regardless of reward sparsity, and Reset is especially worth trying if it is a non-sparse reward task.

### 4.1.3 The impact of neuron dominance

In Figure 9, we present scatterplots that demonstrate the relationships between the largest activation score and three metrics: return, effective rank, and the subsequent layer's dormant percentage. Each data point in the figure shows the result averaged across all tested tasks and all random seeds. As indicated by the results, approaches that manage to keep a low largest activation score (ReDo, Reset, LNWD, and HL-Gauss) tend to have high sranks, high returns, and low dormant percentages, compared to the other two approaches (Baseline and Pruning). This suggests that maintaining a low degree of neuron dominance is beneficial for keeping a high representation capacity and ultimately for improving the performance, which corroborates our theoretical result in Section 3.3. Separate scatterplots for each task are presented in Appendix C.

### 4.1.4 LayerNorm's effect on neuron dominance

Recall that the sparsified Reacher-easy task, the reward is masked to 0 during the first half of training, and is unmasked in the second half. The original Reacher-easy is solvable by all these approaches as demonstrated in the second row of Figure 6. Thus, if the network maintains the same level of plasticity as at initialization, it should rapidly converge to the optimal policy after the rewards become visible. However, as a proof of lost plasticity, we can see in the last row of Figure 6 that once the reward is unmasked, the baseline and Reset fail to learn a meaningful policy (dominant neurons also occur in baseline runs as shown by Figure 2), while ReDo can only learn a suboptimal policy. Pruning oscillates a lot during training. Only HL-Gauss and LNWD manage to converge to the optimal policy.

By definition, LN transforms the preactivation distribution into a zero-centered distribution with unit variance, which means that even if all the preactivations are negative (full saturation), LN is able to revive

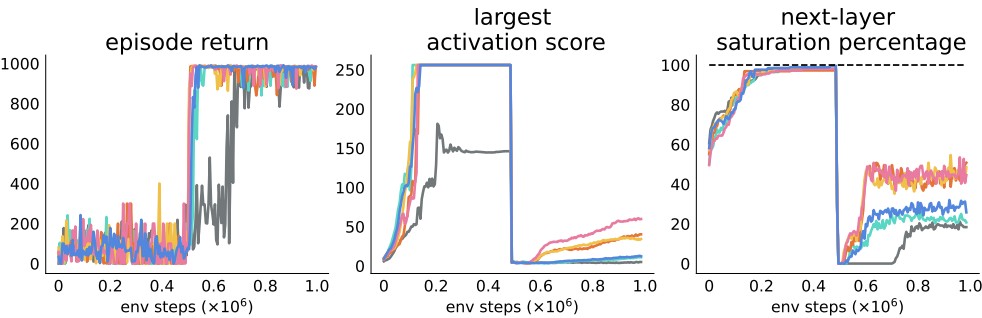

Figure 10: Learning curves on sparsified reacher-easy for DrQ without LN in the first half of training and with LN in the second half of training. Each curve represents one single run with a different random seed. Compared to Figure 2 (d) where the agents use critic networks without LN throughout training, here LN significantly improves the performance, and is able to eliminate dominant neurons and revive a fully saturated layer back to normal. Results on two more reward non-stationary tasks exhibit similar trends (see Appendix C.4).

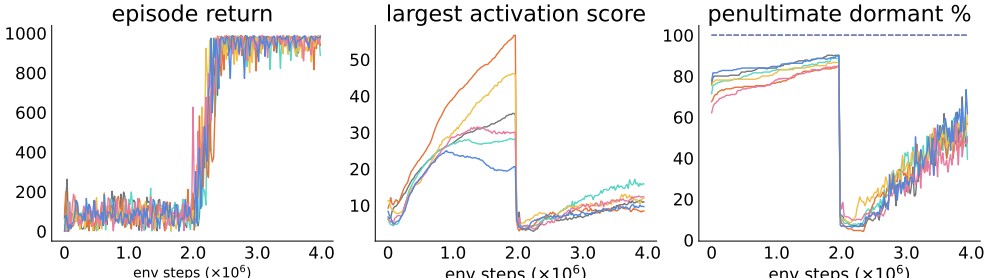

Figure 11: Statistics of HL-Gauss on sparsified reacher-easy. Each curve represents one single run with a different random seed. We can see that HL-Gauss is able to prevent the penultimate layer from being fully saturated, and as long as the reward signal is revealed to the network, the penultimate dormant percentage instantly returns to a low value. Results on two more reward non-stationary tasks exhibit similar trends (see Appendix C.5).

this layer and thus bring the learning dynamics back to normal. We empirically verify this in the sparsified Reacher-easy task, by first using a value network without LN, and then adding LN to the preactivations at the second half of the training process. The results in Figure 10 show that the penultimate layer of DrQ's value network suffers from representation collapse and becomes fully saturated after the first half of training. However, once LN is added, the dominant neurons are eliminated, and the penultimate layer revives immediately. The agent still learns the optimal policy using a small number of samples. Therefore, LayerNorm is a simple yet powerful technique in dealing with neuron dominance as well as saturation. The other component of LNWD, weight decay, instead handles parameter norm growth. We perform an ablation study of this and find that in most tasks using LN alone achieves the same performance as using both two components (see Appendix C for the results). Prior work (Nauman et al., 2024) has also shown that in most DM-Control tasks, parameter norm does not grow drastically and is not problematic.

Notably, compared to LNWD, in sparsified Reacher-easy, HL-Gauss converges to the optimal policy using the same number of samples as in the non-sparse case, suggesting little plasticity loss. Despite its effectiveness, there have been few theoretical results investigating the reasons behind the performance gains achieved by HL-Gauss. In the next subsection, we provide a theoretical understanding of the advantages possessed by classification losses, particularly in the sparse reward setting.

### 4.2 Classification losses prevent representation collapse

Although not directly related to dominant neurons, there is an interesting phenomenon we observe in our experiments that might help explain why classification losses can prevent representation collapse: as shown in Figure 11, even if the agent continues using a zero reward in value learning for a large number of time steps, HL-Gauss never exhibits a 100% saturation in either of the hidden layers, and the saturation rate immediately declines when the agent begins to receive non-zero rewards, without any additional intervention from parameter-perturbation approaches like ReDo. While classification losses do not directly target dominant neurons, their ability to prevent representation collapse suggests a complementary mechanism for preserving plasticity when dominant activations arise. This phenomenon motivates us to investigate the differences between the MSE regression loss and the categorical cross-entropy loss in terms of their representation dynamics.

Consider an MDP comprising a finite state space $\mathcal{X}$, of which $P^\pi \in \mathbb{R}^{|\mathcal{X}| \times |\mathcal{X}|}$ and $R^\pi \in \mathbb{R}^{|\mathcal{X}|}$ are respectively the transition matrix and reward function given $\pi$. Denote $\Psi \in \mathbb{R}^{N \times |\mathcal{X}|}$ as the feature matrix and $W \in \mathbb{R}^N$ as the final-layer weights where $N$ is the width of the penultimate layer. Our analysis is inspired by prior work by Lyle et al. (2021). In the regression case considered in their paper, the goal is to minimize the temporal difference, i.e., $\|R^\pi + \gamma P^\pi \Psi^T W - \Psi^T W\|_2^2$. By assuming a fixed ensemble of linear heads $\{W^m\}_{m=1}^M$ and a specific way of its initialization, they simplify the learning dynamics in the zero-reward setting as:

$$\lim_{M \to \infty} \partial_t \Psi_t^M \overset{P}{=} -\left(I - \gamma P^\pi\right) \Psi_t, \tag{5}$$

which can be analytically solved as:

$$\lim_{M \to \infty} \Psi_t^M \overset{P}{=} \exp\left(-t\left(I - \gamma P^\pi\right)\right) \Psi_0, \tag{6}$$

where $M$ denotes the number of linear heads. Though such linear heads are typically not used in practice, their solution remains insightful by indicating that in the case of zero reward the representation learned by the value network will eventually collapse to zero (for all inputs), i.e., $\lim_{t \to \infty} \lim_{M \to \infty} \Psi_t^M \overset{P}{=} \mathbf{0}$, which matches our observations in Section 3.1. By extending their derivation, we characterize the learning dynamics in the classification case in Theorem 3.

**Theorem 3.** *Denote the number of logits by $G$. Under the condition of zero reward and using one-step temporal difference learning with the categorical cross-entropy loss, the continuous-time dynamics of the representation $\Psi \in \mathbb{R}^{N \times |\mathcal{X}|}$ and the final-layer weights $W_t \in \mathbb{R}^{N \times G}$ are*

$$\partial_t W_t = \alpha \Psi_t \left( (\gamma P^\pi - I) \frac{\exp(\Psi_t^T W_t)}{\exp(\Psi_t^T W_t) \cdot \mathbf{1}_{G \times G}} \right), \tag{7}$$

$$\partial_t \Psi_t = \beta W_t \left( (\gamma P^\pi - I) \frac{\exp(\Psi_t^T W_t)}{\exp(\Psi_t^T W_t) \cdot \mathbf{1}_{G \times G}} \right)^T, \tag{8}$$

*respectively, where $\alpha, \beta$ are learning rates and $\mathbf{1}_{G \times G}$ is a $G \times G$ matrix with all entries being 1.*

Compared to the MSE case, the classification case in Theorem 3 is not directly solvable even with infinite linear heads and under the same assumptions as Lyle et al. (2021), partly because of the non-linear softmax process that converts logits to probabilities. However, under the assumption of a fixed weight $W$, Equation 8 implies that even if $\Psi_t$ collapses to $\mathbf{0}$ at some time step $t$, due to softmax its gradient $\partial_t \Psi_t$ is not going to be $\mathbf{0}$ (as $(\gamma P^\pi - I)$ is full-rank), unlike the case of the MSE loss (Equation 5). Therefore, it is likely that the non-linearity (i.e., corresponding to $\frac{\exp(\Psi_t^T W_t)}{\exp(\Psi_t^T W_t) \cdot \mathbf{1}_{G \times G}}$ in Equation 8) that makes the categorical dynamics intractable may in fact be what helps it avoid representation collapse.

Intuitively, in the case of zero reward, the probability mass will be mostly concentrated on the first bin of the categorical representation as it corresponds to the probability of taking the lowest value, so if the representation $\Psi$ or weight matrix $W$ collapses to a zero matrix then due to the softmax process the output will always be a uniform distribution which does not fit the target. This produces a contradiction since fitting a one-hot target that is fixed for all inputs is in general not a challenging task even for shallow networks.

## 5 Related work

**Understanding plasticity loss** The term "plasticity loss" has been adopted to describe the loss of a network's ability to overwrite its prior predictions in response to new experience (Lyle et al., 2023) and has recently gained a large amount of interest in the deep RL community (Klein et al., 2024). Prior work has leveraged the linear algebraic properties (e.g., spectral norm, rank, etc.) of the learned representations to quantify and analyze the expressivity of the network in deep Q-learning (Kumar et al., 2020; Gogianu et al., 2021; Lyle et al., 2022). Lyle et al. (2023) show that normalization layers and categorical representations, methods considered to smoothen the loss landscape, are identified as the two most effective architectural choices that best preserve plasticity. This matches our experiment results in which HL-Gauss and LNWD perform exceptionally well. Also, our theoretical analysis for categorical losses' resistance to representation collapse verifies the empirical results in Lyle et al. (2023) where two-hot encodings are better at preserving plasticity. From a more microscopic level, Sokar et al. (2023) discover the dormant neuron phenomenon in deep RL, suggesting this as a source of plasticity loss. Our work complements these works by revealing a potential cause for plasticity loss, i.e., the emergence of disproportionally large activations, and discussing their connection to concepts from representation learning, unveiling their impact on network expressivity.

**Dealing with plasticity loss** Various approaches have been proposed to deal with plasticity loss. While some focus on preventing plasticity loss from occurring either by modifying network architectures (e.g., modifying activation functions (Abbas et al., 2023; Park et al., 2025) and adding normalization layers (Lyle et al., 2024a;b; Juliani & Ash, 2025) or regularizing network parameters (Gogianu et al., 2021; Lyle et al., 2022; Lee et al., 2024; Obando-Ceron et al., 2024; Chung et al., 2024), others seek to periodically rejuvenate networks that have lost plasticity (Igl et al., 2020; Dohare et al., 2021; Nikishin et al., 2022; Sokar et al., 2023; Nikishin et al., 2023; Ji et al., 2024). There is also work aiming at preserving plasticity by improving the optimizer (Lee et al., 2023; Muppidi et al., 2025). Instead of using the activation score as the criterion for resetting dormant neurons, Liu et al. (2025) argue that the gradient magnitude is a more reliable metric than the activation. We provide a gradient-based analysis in Appendix C.2, which shows that the gradient becomes zero when neuron dominance occurs, verifying the theoretical intuition. Our work reveals the negative impact of neuron dominance and provides a novel perspective to interpret the effectiveness of existing approaches that deal with plasticity loss. Our results also provide useful practical guidance for employing these approaches in sparse and non-sparse reward tasks. Recently, Qin et al. (2024) identify the "over-active" neuron in multi-agent reinforcement learning. It refers to neurons with an expected activation score larger than a threshold, which resembles the dominant neuron in our paper. However, the threshold in their definition is a pre-defined value, and does not concern layer width. In contrast, we use a threshold that depends on the layer width, which enables the derivation of our theory that dominant neurons result in saturation in the next layer of a neural network.

## 6 Conclusion

This work proposes dominant neurons as a potential cause of network plasticity loss through both theoretical and empirical investigations. To our best knowledge, this is the first work to show that dominant neurons can induce saturation in subsequent layers. We examine this phenomenon through the lens of representation learning and establish a connection to the previously studied representation collapse phenomenon in deep reinforcement learning.

We further study several mitigation strategies by analyzing their effects on neuron dominance and comparing their performance across a variety of visual control tasks. Our results show that approaches which maintain low peak activation scores tend to exhibit greater representational capacity, lower fractions of inactive neurons, and better overall performance. In addition, we provide a theoretical analysis explaining why classification losses exhibit inherent advantages in preventing representation collapse, offering new insight into the effectiveness of approaches such as HL-Gauss. In addition, our results (Figure 7) also indicate that for ReDo and Pruning, intervention costs outweigh the gains in non-sparse reward environments. Although these tasks do exhibit reduced neuron dominance and dormancy, the exact negative impact of ReDo and pruning on learning dynamics is not yet fully understood. We leave the formal characterization of this relationship for future research.

Although our empirical study focuses on visual DM-Control tasks, our analysis applies broadly to ReLU-based value networks and helps explain why widely used techniques such as normalization and classification-based objectives improve stability in deep reinforcement learning. We hope these findings motivate future work on activation-aware diagnostics and mitigation strategies for maintaining plasticity in deep RL.

**Limitations**    Our empirical evaluation of mitigation strategies is conducted exclusively with the DrQ algorithm and its default architecture on the visual DM-Control suite. While this provides a controlled setting for studying neuron dominance, the relative effectiveness of the tested strategies may differ under other algorithms, network architectures, hyperparameter configurations, or domains. In particular, the substantial variation in results across individual tasks suggests sensitivity to task-specific properties. We therefore encourage future work to validate these findings across a broader range of experimental settings.

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

# A Proofs

## A.1 Proof for Theorem 1

**Theorem 1.** *Assume that for any neuron $m$ in layer $k+1$, the weight $w_{1m}^{k+1}$ connecting a dominant neuron in layer $k$ and this neuron is no less than the average of weights from the same layer that connect to this neuron, i.e., $w_{1m}^{k+1} \geq \frac{1}{N_k} \sum_{i=1}^{N_k} w_{im}^{k+1}$, and that the biases in layer $k+1$ are zero. Under these assumptions, a neuron in layer $k+1$ will either be a saturated or a zombie neuron.*

*Proof.* By the assumption we have $\mathbb{E}\left[|w_{1m}^{k+1} g_1^k(x)|\right] > \sum_{j \neq 1} \mathbb{E}\left[|w_{jm}^{k+1} g_j^k(x)|\right]$, and thus

$$
\begin{cases}
\sum_{j=1}^{N_k} \mathbb{E}[w_{jm}^{k+1} g_j^k(x)] > 0 & \text{if} \quad w_{1m}^{k+1} > 0, \\
\sum_{j=1}^{N_k} \mathbb{E}[w_{jm}^{k+1} g_j^k(x)] \leq 0 & \text{otherwise.}
\end{cases}
$$

Thus, the sign of the preactivation of a penultimate (layer $k+1$) neuron $m$, depends in expectation on the sign of the outgoing weight of the dominant neuron, i.e., $w_{1m}^{k+1}$. $\qquad\square$

**Remark:** The zero-bias assumption in layer $k+1$ can be relaxed to $b_m \leq \mathbb{E}\left[|w_{1m}^{k+1} g_1^k(x)|\right] - \sum_{j \neq 1} \mathbb{E}\left[|w_{jm}^{k+1} g_j^k(x)|\right]$.

## A.2 Proof for Theorem 2

**Theorem 2.** *Denote $\hat{s}_i$ as the sum of the $i$-th column of the sampled feature matrix $\hat{\Phi} \in \mathbb{R}^{B \times N}$, where $N$ is the layer width and $B$ is the batch size. Denote the singular values of the sampled matrix as $\{\hat{\sigma}_i\}$. Assume that the samples are i.i.d. and let $n = \min(N, B)$. Assuming, without loss of generality, that the index of the largest neuron is 1 (i.e., $\mathbb{E}[s_1] \geq \mathbb{E}[s_j], \forall j$), we have:*

$$\mathbb{E}\left[\frac{\hat{\sigma}_1}{\sum_{i=1}^N \hat{\sigma}_i}\right] \geq \sqrt{\frac{1}{nB} + \frac{2(B-1)}{n\left[B(1 + Var(\hat{s}_i)) + \frac{\sum_{i=2}^N \mathbb{E}[\hat{s}_i^2]}{(\mathbb{E}[\hat{s}_1])^2}\right]}}. \tag{4}$$

*Proof.* According to the min-max theorem for singular values, $\sigma_1 = \max_{\|y\|_2=1} \|yX\|_2$. Let $y^T = (\frac{1}{\sqrt{B}}, \ldots, \frac{1}{\sqrt{B}})^T \in \mathbb{R}^B$ so that $\|y\|_2 = 1$, then we have:

$$\|yX\|_2^2 = \frac{1}{B} \sum_{i=1}^N s_i^2, \tag{9}$$

and so $\sigma_1 \geq \sqrt{\frac{1}{B} \sum_{i=1}^N s_i^2}$. According to Cauchy-Schwarz inequality, $\sum_{i=1}^n \sigma_i \leq \sqrt{n \cdot \sum_{i=1}^n \sigma_i^2}$. Also, by the definition of singular values we have $\sum_{i=1}^n \sigma_i^2 = \text{tr}(X^T X) = \sum_{i,j} x_{ij}^2$, and hence $\sum_{i=1}^n \sigma_i \leq \sqrt{n \cdot \sum_{i,j} x_{ij}^2}$. Putting the above two inequalities together there is:

$$\frac{\sigma_1}{\sum_{i=1}^n \sigma_i} \geq \sqrt{\frac{\sum_{i=1}^N s_i^2}{nB \sum_{i,j} x_{ij}^2}}. \tag{10}$$

To derive a more interpretable result, we further scale it:

$$\frac{\sum_{i=1}^N s_i^2}{\sum_{i,j} x_{ij}^2} \geq \frac{\sum_{i=1}^N s_i^2}{\sum_j x_{j1}^2 + \sum_{i=2}^N s_i^2} \tag{11}$$

$$= \frac{(\sum_j x_{j1})^2 + \sum_{i=2}^N s_i^2}{\sum_j x_{j1}^2 + \sum_{i=2}^N s_i^2} \tag{12}$$

$$= 1 + \frac{2 \sum_{i \neq j} x_{i1} x_{j1}}{\sum_j x_{j1}^2 + \sum_{i=2}^N s_i^2}. \tag{13}$$

Since the samples are i.i.d., for entries in the same row $i$ we have $\mathbb{E}[x_{ij}] = \mathbb{E}[x_{ik}] = \frac{1}{B}\mathbb{E}[s_i]$. Also, when $j \neq k$, $\mathbb{E}[x_{ij}x_{ik}] = \mathbb{E}[x_{ij}]\mathbb{E}[x_{ik}] = \frac{1}{B^2}(\mathbb{E}[s_i])^2$, and $\mathbb{E}[x_{ij}^2] = (\mathbb{E}[s_i])^2 + Var(s_i)$, so:

$$\mathbb{E}\left[\frac{2 \sum_{i \neq j} x_{i1} x_{j1}}{\sum_j x_{j1}^2 + \sum_{i=2}^N s_i^2}\right] \geq \frac{2\mathbb{E}[\sum_{i \neq j} x_{i1} x_{j1}]}{\mathbb{E}[\sum_j x_{j1}^2 + \sum_{i=2}^N s_i^2]} \quad \text{\small (Due to Jensen's inequality)} \tag{14}$$

$$\geq \frac{2B(B-1)(\mathbb{E}[s_1])^2}{B \cdot ((\mathbb{E}[s_1])^2 + Var(s_1)) + \sum_{i=2}^N \mathbb{E}[s_i^2]} \tag{15}$$

$$\geq \frac{2B(B-1)}{B(1 + Var(s_i)) + \frac{\sum_{i=2}^N \mathbb{E}[s_i^2]}{(\mathbb{E}[s_1])^2}}. \tag{16}$$

Substituting into equation 10, we have:

$$\mathbb{E}\left[\frac{\sigma_1}{\sum_{i=1}^B \sigma_i}\right] \geq \sqrt{\frac{1}{nB} + \frac{2(B-1)}{n\left[B(1 + Var(s_i)) + \frac{\sum_{i=2}^N \mathbb{E}[s_i^2]}{(\mathbb{E}[s_1])^2}\right]}}. \tag{17}$$

$$\square$$

## A.3 Proof for Theorem 3

**Theorem 3.** *Denote the number of logits by $G$. Under the condition of zero reward and using one-step temporal difference learning with the categorical cross-entropy loss, the continuous-time dynamics of the representation $\Psi \in \mathbb{R}^{N \times |\mathcal{X}|}$ and the final-layer weights $W_t \in \mathbb{R}^{N \times G}$ are*

$$\partial_t W_t = \alpha \Psi_t \left( (\gamma P^\pi - I) \frac{\exp(\Psi_t^T W_t)}{\exp(\Psi_t^T W_t) \cdot \mathbf{1}_{G \times G}} \right), \tag{7}$$

$$\partial_t \Psi_t = \beta W_t \left( (\gamma P^\pi - I) \frac{\exp(\Psi_t^T W_t)}{\exp(\Psi_t^T W_t) \cdot \mathbf{1}_{G \times G}} \right)^T, \tag{8}$$

*respectively, where $\alpha, \beta$ are learning rates and $\mathbf{1}_{G \times G}$ is a $G \times G$ matrix with all entries being 1.*

*Proof.* In the following we vectorize the notations over the whole state space, denoted by bold characters. For instance, $\boldsymbol{l^i} := [l^i(x_1), \ldots, l^i(x_{|\mathcal{X}|})]^T$. Denote $\boldsymbol{x}$ as the state vector, $\boldsymbol{l} = [\boldsymbol{l^1}, ..., \boldsymbol{l^G}] = \check{\Psi}^T W$ as the logits where $\boldsymbol{l^i} \in \mathbb{R}^{|\mathcal{X}|}$, $\Psi \in \mathbb{R}^{N \times |\mathcal{X}|}$ as the representation vector, $W \in \mathbb{R}^{N \times G}$ as the weight matrix of the final layer, assuming $G$ logits and $N$ penultimate layer width. Denote the label probability as $\boldsymbol{p} = [\boldsymbol{p^1}, ..., \boldsymbol{p^G}]$, network prediction $\hat{\boldsymbol{p}} = \text{Softmax}(\boldsymbol{l}) = [\hat{\boldsymbol{p}}^1, \ldots, \hat{\boldsymbol{p}}^G]$ where $\hat{\boldsymbol{p}}^i = \frac{\exp(\boldsymbol{l^i})}{\sum_j \exp(\boldsymbol{l^j})} \in \mathbb{R}^{|\mathcal{X}|}$. Denote $e_i$ as the $i$-th unit vector in $\mathbb{R}^G$.

For an arbitrary single state, the continuous-time learning dynamics of the weight matrix and the feature are:

$$\partial_t W_t = -\alpha \nabla_{W_t} \left( -\sum_{i=1}^{G} \text{SG}[p_t^i] \log \hat{p}_t^i \right) = \alpha \sum_{i=1}^{G} \psi_t p_t^i (e_i^T - \hat{p}_t^T), \tag{18}$$

$$\partial_t \psi_t = -\beta \nabla_{\psi_t} \left( -\sum_{i=1}^{G} \text{SG}[p_t^i] \log \hat{p}_t^i \right) = \beta \sum_{i=1}^{G} p_t^i (w_t^i - W_t \hat{p}_t), \tag{19}$$

respectively, where $\text{SG}$ denotes "stop gradient", $\psi_t \in \mathbb{R}^N$ denotes the representation of the state, $W = [w^1, \ldots, w^G], w^i \in \mathbb{R}^N$, $\hat{p}_t \in \mathbb{R}^G$ is the prediction probabilities for this state, $p_t^i$ is a scalar representing the $i$-th label for this state, and $\alpha, \beta$ are learning rates. Now by vectorizing these two equations over the entire state space, we have:

$$\partial_t W_t = -\alpha \nabla_{W_t} \left( -\sum_{i=1}^{G} \text{SG}[\boldsymbol{p}_t^i] \circ \log \hat{\boldsymbol{p}}_t^i \right) \tag{20}$$

$$= \alpha \sum_{i=1}^{G} \Psi_t \text{diag}(\boldsymbol{p}_t^i)([e^i, \ldots, e^i]^T - \hat{\boldsymbol{p}}_t) \tag{21}$$

$$= \alpha \Psi_t (\boldsymbol{p}_t - \hat{\boldsymbol{p}}_t), \tag{22}$$

where $\circ$ denotes element-wise product, $E^i = \mathbf{1}_{|\mathcal{X}|} \cdot e_i^T, W_t^i = w_t^i \cdot \mathbf{1}_{|\mathcal{X}|}^T$, and $\text{diag}(\boldsymbol{p}_t^i) = \text{diag}(p_t^i(x_1), \ldots, p_t^i(x_{|\mathcal{X}|}))$. The weight vector and the unit vector are copied by $|\mathcal{X}|$ times since they are invariant of the state. Similarly, we can derive $\partial_t \Psi_t = \beta W_t (\boldsymbol{p}_t^T - \hat{\boldsymbol{p}}_t^T)$.

Assuming $R^\pi = 0$, to obtain $\boldsymbol{p}_t$ we do not have to transform $\hat{\boldsymbol{p}}_t$ into $V_t$ and then re-transform it back to probabilities like what is done in HL-Gauss. Instead, the transition matrix $P^\pi$ can be directly applied to $\hat{\boldsymbol{p}}$, and so:

$$\partial_t W_t = \alpha \Psi_t (\boldsymbol{p}_t - \hat{\boldsymbol{p}}_t) = \Psi_t \left( (\gamma P^\pi - I) \frac{\exp(\Psi_t^T W_t)}{\exp(\Psi_t^T W_t) \cdot \mathbf{1}_{G \times G}} \right), \tag{23}$$

$$\partial_t \Psi_t = \beta W_t (\boldsymbol{p}_t - \hat{\boldsymbol{p}}_t)^T = W_t \left( (\gamma P^\pi - I) \frac{\exp(\Psi_t^T W_t)}{\exp(\Psi_t^T W_t) \cdot \mathbf{1}_{G \times G}} \right)^T. \tag{24}$$

$\square$

# B Experiment details

## B.1 Modification of the reward functions in 3 tasks

Since in the original DM-Control suite only Cartpole-swingup_sparse has a reward function that is sufficiently sparse to induce completely saturated hidden layers, we modify 4 more tasks to demonstrate this phenomenon. Here we detail the modifications in the code level. All the modifications below are conducted in corresponding files under https://github.com/google-deepmind/dm_control/tree/main/dm_control/suite.

Cartpole-balance_sparse: We set

- `_CART_RANGE = (-.0001, .0001)`
- `_ANGLE_COSINE_RANGE = (.9999, 1)`

Finger-spin: We set

- `_INITIAL_SPIN_VELOCITY = 10`
- `_STOP_VELOCITY = 20.0`
- `_SPIN_VELOCITY = 30.0`

Reacher-easy: we intend to simulate the long-term zero-reward condition in the Theorem of representation collapse, and then test the networks' resiliency after the zero-reward phase. To achieve this, we mask the reward in critic updates for the first 500k gradient steps, and then unveil it after this period.

Note that all the neurons in the first hidden layer become saturated following the emergence of a dominant neuron, resulting in NaNs in score computing. To avoid confusion in reading, the NaNs in Figure 2 are replaced with the last non-Nan value in the data sequence.

## B.2 Hyperparameter settings

We adopt the default network architecture of DrQ which consists of a shared 4-layer CNN encoder plus two 3-layer MLP heads for the actor and the critic, respectively. To accelerate the computation, we use a Jax (Bradbury et al., 2018) version of the implementation [3]. The general hyperparameter settings follow the default setting in this implementation: for all tested tasks, the discount factor is 0.99, the learning rate is 0.0003 for both actor and critic, the batch size is 512, the $\tau$ value for soft target network update is 0.005, and the buffer size is 100000. The optimizer is Adam with default hyperparameters (Kingma & Ba, 2014), i.e., betas=(0.9, 0.999), eps=1e-08, no weight decay.

**Reset** The official implementation resets both the actor and critic heads of the DrQ network, here we only reset the critic head for a fair comparison with methods that exclusively target the value network such as HL-Gauss. We find the performance comparable to resetting both heads. We tune the reset interval within the following values: $10k, 100k, 1000k$, and adopt $100k$ (same as the default value) as the final setting.

**ReDo** We transfer the official implementation in Atari to the DM-Control domain and keep the core functions implementing ReDo unchanged. For the dormant threshold we adopt the default 0.1 in their implementation. We tune the interval for resetting these neurons within the following values: $100, 1k, 10k$, and adopt $1k$ (same as the default value) as the final setting.

**Gradual Magnitude Pruning** We adopt the official implementation and use the recommended target sparsity of 0.95 and pruning schedule which starts at 20% and ends at 80% of training. We tune the interval for pruning within the following values: $100, 1k, 10k$, and adopt $1k$ (same as the default value) as the final setting.

---

[3]https://github.com/ikostrikov/jaxrl

**LNWD**   The weight decay parameter is set to the default 0.0001 in our experiments.

**HL-Gauss**   We use 101 logits (number of bins) and set $\sigma$ to 0.75 in our experiments [4] For the value interval, we use $[0, 100]$ which is exactly the ground-truth value interval for all DM-Control tasks Tassa et al. (2018) [5] We find that in some non-sparse reward tasks where HL-Gauss achieves low return (e.g., Cheetah-run), increasing the maximum value [6] and $\sigma$ by the same scale significantly improves the performance (e.g., setting the maximum value to 1000 and $\sigma$ to 7.5). However, in other tasks this appears to be detrimental. As such, we adopt $[0, 100]$ for the value interval and 0.75 for $\sigma$ across all tested tasks.

### B.3   CIFAR10 experimental details

We train a convolutional classifier on CIFAR-10 ($32 \times 32 \times 3$ RGB). The backbone consists of four $3 \times 3$ convolutional layers with channel sizes $(64, 128, 256, 256)$. The first convolution uses stride 2; the remaining three use stride 1. All convolutions use same padding and no bias terms. We apply global average pooling over the spatial dimensions. The pooled feature vector is processed by a two-layer multilayer perceptron (MLP) with hidden width 512 and ReLU activations, followed by a linear layer mapping to 10-dimensional logits (one per class). The optimizer is Adam and the learning rate is 0.005. The incoming weights of the largest neuron in the first MLP layer is multiplied by 1e7 when the validation accuracy reaches 0.5.

## C   Additional results

### C.1   Neuron dominance in DrQ-ResNet18

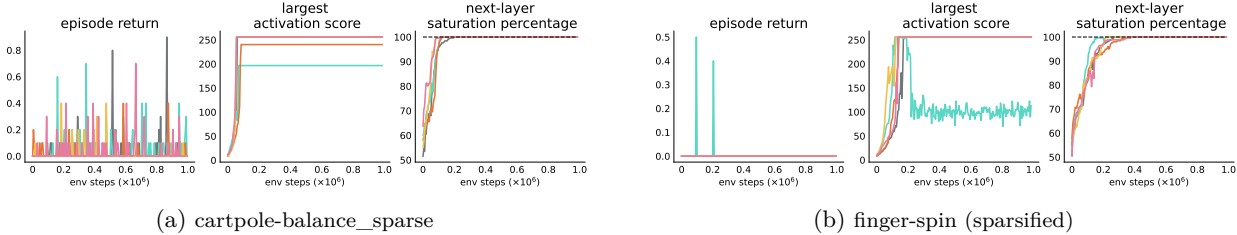

(a) cartpole-balance_sparse                              (b) finger-spin (sparsified)

Figure 12: Results of individual runs of DrQ-ResNet18 on 2 sparse reward tasks. DrQ-ResNet18 replaces the CNN layers in the original DrQ-v1 implementation with ResNet-18 (He et al., 2016).

As shown by Figure 12, the phenomenon consistently occurs when the CNN layers are replaced by another mainstream architecture, ResNet-18 (He et al., 2016; Sokar et al., 2023; Obando-Ceron et al., 2024).

### C.2   Gradient-based analysis

Liu et al. (2025) show that a gradient-based metric resets neurons more reliably than the activation-based metric used in ReDo (Sokar et al., 2023). From the gradient perspective, dominant neurons cause next-layer saturation, and in dense layers with ReLU activation, this means that the gradient to the layer containing the dominant neuron will become zero. Figure 13 verifies this intuition by showing that the gradient converges to 0 when neuron dominance occurs.

### C.3   Scatter plots for correlations between srank, max score, and dormant percentage

Here we present scatterplots for each task separately in Figure 14, corresponding to the aggregated plot in Figure 9 in the main paper.

---

[4]We use the code provided by Farebrother et al. (2024) for experiments.

[5]The reward for each step is between 0 and 1, and there is 1000 steps each episode. Therefore, the minimum value is 0, and the maximum value is $\sum_{t=0}^{999} \gamma^t \cdot 1 \approx 99.996$ given $\gamma = 0.99$.

[6]The upper bound of the value interval which is then divided to discrete bins for using classification losses.

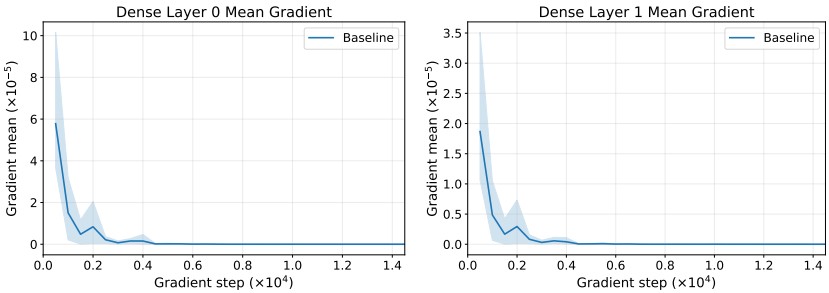

(a) Mean gradient magnitude to the layer that contains the dominant neuron.

(b) Mean gradient magnitude to the subsequent layer of the dominant neuron.

Figure 13: Gradient magnitude of the MLP layers in Cartpole-balance_sparse, corresponding to the failing runs in Figure 2(c). The gradient converges to 0 over the course of training.

### C.4   LayerNorm can revive a fully saturated layer

Figure 16 shows that applying LayerNorm to a fully saturated network can rejuvenile the network.

### C.5   Performance of HL-Gauss on reward non-stationary tasks

Figure 17 shows that HL-Gauss prevents the penultimate layer from being fully saturated when the agent keeps receiving zero reward, and as long as the reward signal is revealed to the network, the penultimate dormant percentage instantly returns to a low value.

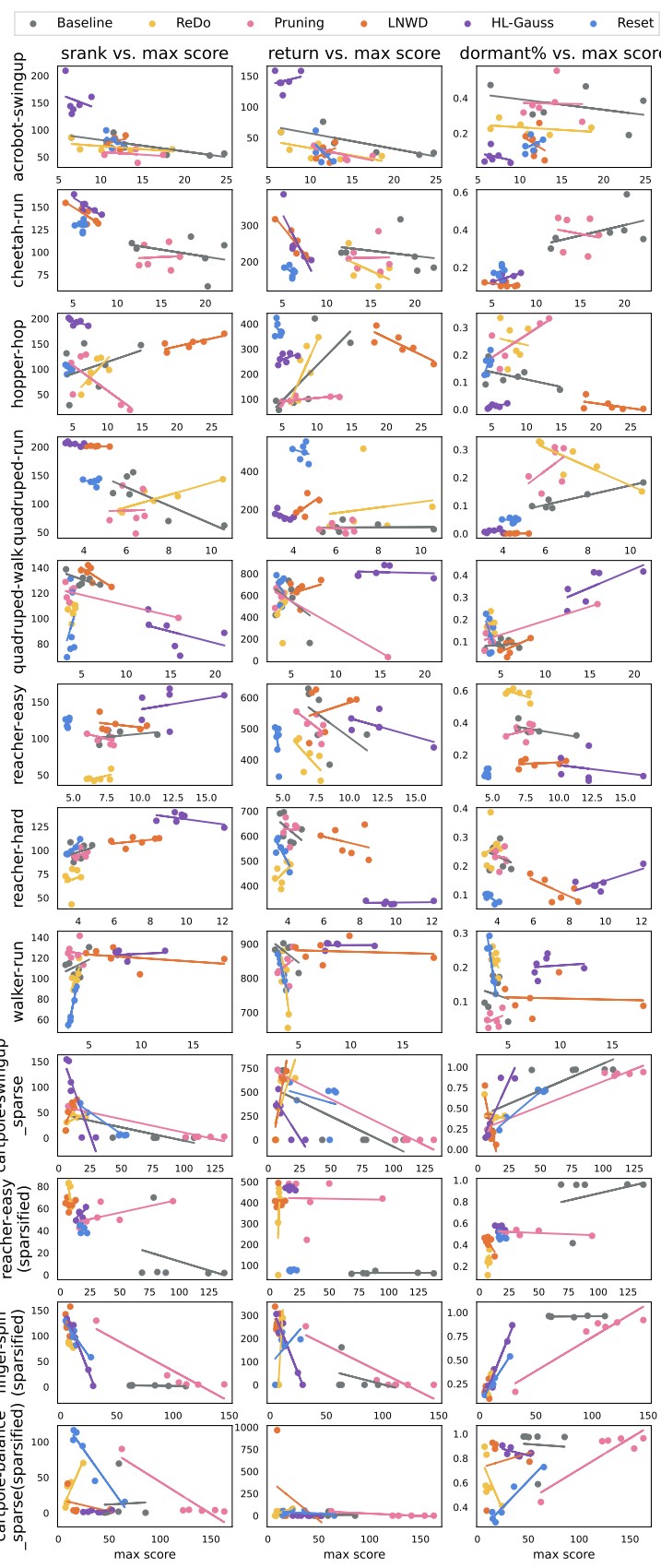

Figure 14: Results showing the correlation between the largest activation score and three metrics: srank, return, and dormant percentage, respectively. Each data point represents one single run.

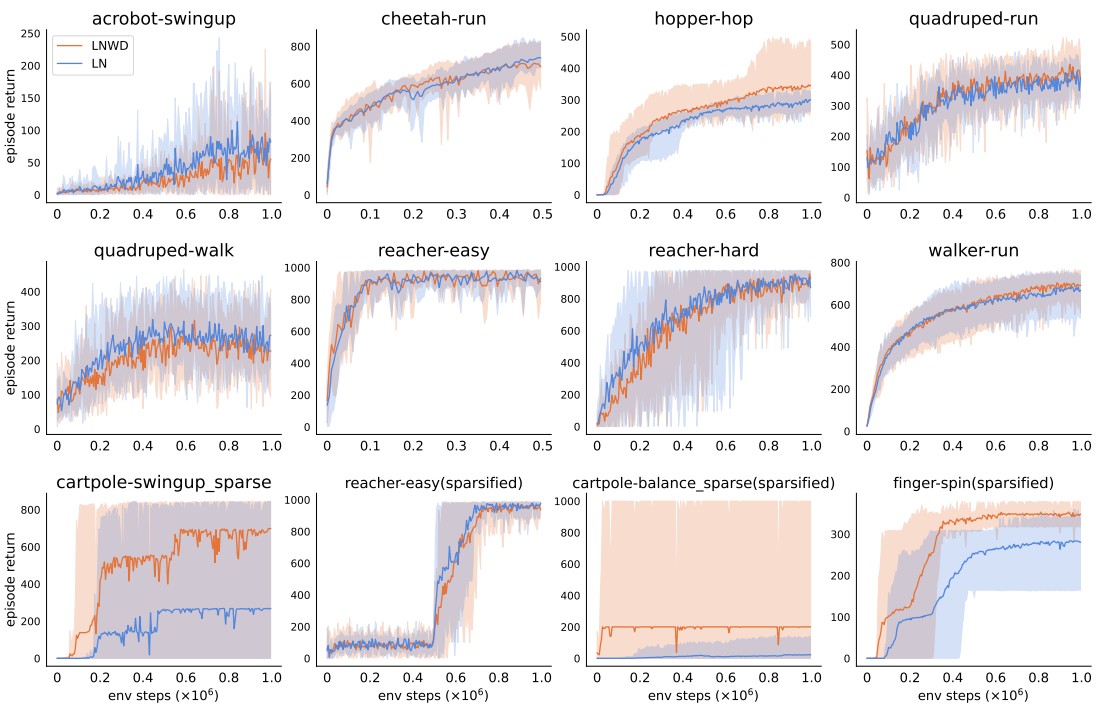

Figure 15: Comparison between LN (LayerNorm alone) and LNWD (LayerNorm with Weight Decay). Weight Decay does not have a significant influence in 9 out of 12 tasks, and is only beneficial in 3 sparse-reward tasks.

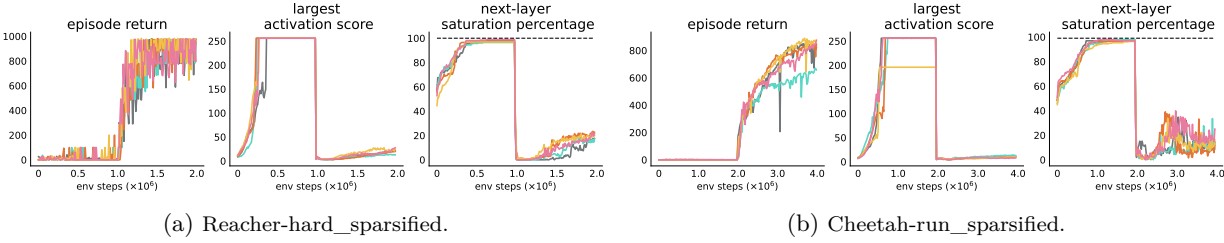

(a) Reacher-hard_sparsified.

(b) Cheetah-run_sparsified.

Figure 16: LayerNorm is able to eliminate dominant neurons and revive a fully saturated layer back to normal.

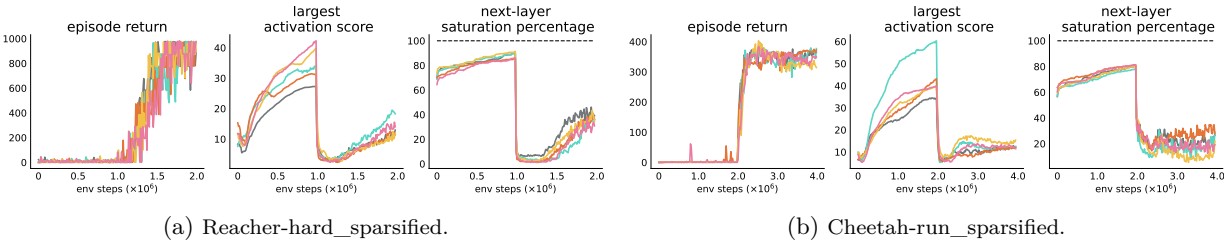

(a) Reacher-hard_sparsified.

(b) Cheetah-run_sparsified.

Figure 17: HL-Gauss is able to prevent the penultimate layer from being fully saturated, and as long as the reward signal is revealed to the network, the penultimate dormant percentage instantly returns to a low value.

