# OpenReview forum: "Understanding the Effects of Neuron Dominance in Deep Reinforcement Learning"
_TMLR — Accepted by TMLR_

### Review · Reviewer_rDrj · 2026-03-10

**Summary Of Contributions:**

This paper investigates the phenomenon of "dominant neurons"—neurons with disproportionately large activation values—and their impact on the performance and plasticity of deep reinforcement learning (RL) agents. While prior work has focused on "dormant neurons" (inactive neurons) as a source of capacity loss, this work identifies neuron dominance as a primary driver that *induces* dormancy in subsequent layers.

**Key Contributions:**
1.  **Identification and Formalization:** The authors provide a formal definition of dominant neurons based on activation scores relative to layer width.
2.  **Theoretical Analysis:** They provide three key theorems: (a) showing how a dominant neuron in one layer can force saturation (dormancy) in the next, (b) linking neuron dominance to a reduction in the effective rank (srank) of the representation, and (c) characterizing why classification-based losses (like HL-Gauss) are more resistant to representation collapse than MSE.
3.  **Empirical Evaluation:** The paper evaluates several mitigation strategies (Reset, ReDo, Pruning, HL-Gauss, and LayerNorm with Weight Decay) across 12 visual DM-Control tasks.
4.  **Actionable Insights:** They demonstrate that LayerNorm with Weight Decay (LNWD) and classification losses (HL-Gauss) are highly effective at suppressing dominant neurons and maintaining network plasticity.

**Strengths:**
*   **Novel Perspective:** It moves the conversation from the *symptom* (dormant neurons) to a potential *cause* (dominant neurons), providing a more mechanistic understanding of plasticity loss.
*   **Strong Theoretical Grounding:** The connection between dominance and rank collapse is mathematically supported and empirically verified.
*   **Practicality:** The findings explain why common tricks (LayerNorm, classification losses) work well in RL, beyond just "smoothing the landscape."

**Weaknesses:**
*   **Activation-Centric View:** The analysis focuses almost entirely on activation magnitudes. It lacks an exploration of how these dominant neurons affect the gradient flow and optimization dynamics directly.
* **La**

**Audience:**

Yes

**Audience Explanation:**

Plasticity loss and "dead" neurons are currently high-priority topics in the deep RL community. Researchers working on optimization, representation learning, and the stability of value-based methods will find the insights regarding dominant neurons and the theoretical advantages of classification losses over regression highly relevant.

**Broader Impact Concerns:**

No concerns

**Claims And Evidence:**

Yes

**Claims Explanation:**

*   **Empirical Correlation:** Figure 2 and Figure 4 clearly show the co-occurrence of "max activation spikes" and "next-layer saturation" in sparse reward tasks.
*   **Theoretical Proof:** Theorem 1 provides a clear mathematical explanation for the observed empirical correlation.
*   **Ablation and Comparison:** The comparison between LN and LNWD (Figure 11) and the scatter plots (Figure 7) provide convincing evidence that managing activation magnitude is directly tied to maintaining representation rank and performance.

**Requested Changes:**

**Critical for Acceptance:**
1.  **Gradient-Based Analysis:** While the activation-based analysis is clear, the optimization of the network is ultimately driven by gradients. The authors should incorporate an analysis (either theoretical or empirical) of how dominant neurons affect the gradient distribution.
*   *Specifically:* Does a dominant neuron cause the gradients of the weights in the subsequent layer to vanish (due to ReLU saturation) or become biased toward specific directions?
*   *Reference:* Please consider and analyze the perspective from Gardiant [1]. For example, comparing the "Dominant Neuron" metric with gradient-based metrics of neuron utility would significantly strengthen the paper’s claims about model optimization.

[1]: Measure Gradients, Not Activations! Enhancing Neuronal Activity in Deep Reinforcement Learning. NeurIPS 2025
**Strengthening the Work:** Recent studies have found that a metric based on gradient magnitude resets neurons more reliably than the ratio of dormant neurons. Is this an advantage gained by capturing and resetting dominant neurons? Please analyze through experiments or theory.

All the experiments in this paper are mainly based on the DrQ baseline and lack the analysis of policy gradient algorithms. It is suggested to add PPO-related experiments to further support the conclusions.

---

> ### Author Response · Authors · 2026-04-06
>
> We thank the reviewer for the constructive feedback. Modifications reflecting the following responses have been made in the revised paper, highlighted in blue.
>
> #
>
> 1. Gradient-based analysis: We thank the reviewer for recommending a related work. We have added the discussion on this work to our Related Work section, and also added the following analysis to Appendix C.2 in the revised paper.
>
>    - Theoretically: Dominant neurons cause next-layer saturation, and in dense layers with ReLU activation, this means that the gradient to the layer containing the dominant neuron will become zero.
>    - Empirically: Our empirical results show that the gradient becomes zero when dominant neurons occur.
>
> 2. Does the gradient-based neuron resetting strategy gain its advantage by resetting dominant neurons?
>
>    In traditional MLP+ReLU architectures, gradient-based neuron resetting (ReGraMa) is equivalent to activation-based neuron resetting (ReDo), as proved in [1]. Our experiments are based on DrQ-v1, which also uses this network structure for the critic heads. Therefore, ReGraMa does not capture dominant neurons, but identifies dormant neurons instead.
>
> 3. PPO-related analysis:
>
>    We appreciate the reviewer's interest in broader applicability. That said, on-policy methods like PPO are generally not considered competitive baselines for visual DMControl due to their poor sample efficiency in pixel-based continuous control---a well-documented limitation in the visual RL literature (e.g., RAD [2]). Prior work [1,3,4] on plasticity loss/neuron dormancy also exclusively feature off-policy algorithms. While it is theoretically possible to combine representation learning techniques with PPO, doing so would require a substantially different algorithmic design and is beyond the scope of this work. We believe our experiments based on the widely-adopted off-policy baseline (DrQ-v1) provides a meaningful evaluation. To ensure the robustness of our findings, we have added an experiment replacing the CNN layers in DrQ-v1 with ResNet-18 (see Appendix C.1). The same neuron dominance phenomenon is also observed in this architecture.
>
>
> **References**
>
> [1] Liu et al., "Measure gradients, not activations! Enhancing neuronal activity in deep reinforcement learning", 2025.
>
> [2] Laskin et al., "Reinforcement learning with augmented data", 2020.
>
> [3] Sokar et al., "The dormant neuron phenomenon in deep reinforcement learning", 2023.
>
> [4] Obando-Ceron et al., "In value-based deep reinforcement learning, a pruned network is a good network", 2024.

---

> > ### Comment · Reviewer_rDrj · 2026-04-23
> > **Thank you for the reply**
> >
> > Thank you for the refinement of your submission, I believe the claims are now well-supported. It is recommended to include your detailed discussion of related work in the new edition so that readers can distinguish between them.

---

### Review · Reviewer_oc3b · 2026-03-15

**Summary Of Contributions:**

The authors study the phenomenon of dominant neurons as they relate to plasticity in deep reinforcement learning.  They connect these neurons to previously-studied ideas such as dormant neurons, zombie neurons, representation collapse, etc.

They offer several empirical and theoretical analyses.  They also analyze HL-Gauss, which is known to mitigate plasticity problems, through the lens of their analyses.

**Audience:**

Yes

**Audience Explanation:**

The community is quite interested in plasticity and related topics.

**Claims And Evidence:**

No

**Claims Explanation:**

This is a borderline paper in my opinion, and I may swing to acceptance, depending on the discussion and promised changes.

**Questions and major concerns:**

Many hyperparameter settings seem to be missing (for example, optimizer settings, discount factor, etc.).

4.1.1-4.1.2 and claim 3 (overall performance):
- The way the results vary drastically between environments is concerning (they could indicate, eg, that different hyperparameters would change results significantly).  For example, for many of the algorithms that showed little-to-no-improvement for the cartpole-balance_sparsified environment, it seems likely that different hyperparameters would significantly change the result (not just the searched hyperparameters given in the Appendix, but all the missing hyperparameters noted above).  I think the paper would be much stronger without these problems, and/or by simply toning down the claims about mitigation strategies, especially claim 3.
- The fact that the experiments are limited to DrQ (algorithm, architecture, and maybe the default hyperparameters) and DM-Control, makes me concerned about the scope of the claims, especially claim 3 (that is, the claims in 4.1.1 and 4.1.2). I do not think the authors necessarily need to expand to other algorithms/architectures/environment types.  However, if not, I would be more comfortable recommending acceptance if the claims were toned down in accordance with these limitations, and these limitations were better highlighted.

Why use 11 stationary environments and 1 non-stationary environment, and then mix the results together (eg, Figures 3, 5, and 6)?  This seems like questionable experimental design.  (There is a related clarity problem here, see “Requested Changes” below, so please correct me if my understanding is wrong.)

Related point: some of the non-stationary results and discussion, such as those from Figures 8-9, were very compelling and interesting. However, it’s concerning that these were all limited to a single environment, even while there were 11 other stationary environments. The paper could be improved considerably by addressing this issue—perhaps focusing less on the experiments/claims in 4.1.1 and 4.1.2 (deemphasizing those claims and/or improving the experiments), and focusing more on the non-stationary experiments/claims.

I am concerned that the links between activation scores, srank, dormancy/dominance, and plasticity/returns are quite correlational.  For example, Figure 7 is one of the key links here, but it’s very shaky as far as evidence goes (only 6 data points, no strong and clear clear trend, the concerns about the limitations above, etc.).  The theory contributions are arguably another link—these contributions are nice—but the assumptions are too strong for the theory to alleviate this concern.

**Requested Changes:**

Some major possible changes/concerns/questions are noted above; changes that mitigate those concerns are the most important changes. Below, I provide some additional feedback; the problems below should be addressed, but these are minor issues:

Intuition about zombie/saturated neurons is not provided, making the paper slightly harder to read. For example, something like the following would be helpful for zombie neurons (quote from cited paper):
“One such pathology is the linearization of a unit, whereby the unit acts as a linear (or nearly linear) transformation of its inputs, effectively reducing the expressivity of the network. For example, a ReLU unit with only positive inputs will behave like the identity, or any smooth activation unit with very low variance inputs will behave close to a linear function… [Zombie units] can propagate gradients (and indeed permit “perfect” signal propagation); however, the presence of too many of them will reduce the effective expressive power of the network”

The N_k in the denominator of the activation score was initially unintuitive for me.  Consider providing more intuition to help the reader.  My intuition is that this results in an average neuron in a given layer having an activation score of 1, but that was not immediately obvious to me (and perhaps that is the incorrect or incomplete intuition).

The use of DrQ (built on SAC) as the learning algorithm is stated in Section 3 (focused mainly on theory), but not mentioned again.  Section 4 (focused more on the empirical results) mentions the use of the DrQ architecture/network, but does not mention the algorithm, which makes this information hard to find when studying section 4.  Suggested fix: in Section 4, change “DrQ network” to something like “DrQ algorithm and architecture” or “DrQ implementation”.

It’s not clear until 4.1.4 that the sparsified Reacher-easy task is non-stationary, and that the 11 other environments are stationary. This detail should be made clear before 4.1.1. However, there is a bigger concern here, see above.

---

> ### Author Response · Authors · 2026-04-06
>
> We thank the reviewer for the constructive feedback. Modifications reflecting the following responses have been made in the revised paper, highlighted in blue.
>
> #
>
> 1. We follow the default hyperparameter setting in https://github.com/ikostrikov/jaxrl and have added more detailed description in Appendix B.2.
>
> 2. We thank the reviewer for the advice. We have toned down the claims in the following sections:
>
>    1. We have added the following paragraph to Section 4.1.1:
>
>       "We note that performance varies considerably across individual tasks (see Figure 5), and that these results are obtained using a single algorithm (DrQ) and architecture on the DM-Control suite. Different hyperparameter configurations or algorithmic choices could plausibly alter the relative ordering of these approaches. That said, within our experimental setup, the pattern tentatively suggests that approaches with a more substantial impact on the learning dynamics may be more effective than fine-grained parameter modifications at addressing the pathological optimization dynamics studied here."
>
>    2. We have added the following paragraph to Conclusion:
>
>       "**Limitations**  Our empirical evaluation of mitigation strategies is conducted exclusively with the DrQ algorithm and its default architecture on the visual DM-Control suite. While this provides a controlled setting for studying neuron dominance, the relative effectiveness of the tested strategies may differ under other algorithms, network architectures, hyperparameter configurations, or domains. In particular, the substantial variation in results across individual tasks suggests sensitivity to task-specific properties. We therefore encourage future work to validate these findings across a broader range of experimental settings."
>
> 3. Concerns about the correlational links: We have added an experiment in Section 3.2 to strengthen the causal relationship. Specifically, we artificially construct a dominant neuron during training and observe a completely saturated subsequent layer shortly after the dominant neuron occurs. The experiment is conducted in supervised learning because the activation score is partly determined by the data distribution which keeps changing during online RL.
>
> 4. We have added the following paragraph in Section 2 to make the definition of zombie/saturated neurons more intuitive:
>
>    "According to the chain rule, saturated neurons induce a vanishing gradient pathology, effectively decoupling the objective function from the parameter updates in early layers and leading to a precipitous decline in the network's functional expressivity. In contrast, becoming a zombie neuron signifies the "linearization" of the neuron, a state where a unit functions as a linear (or quasi-linear) operator on its inputs. This occurs, for instance, when a ReLU gate is restricted to positive inputs---rendering it an identity mapping---or when smooth activations encounter low-variance inputs, causing them to approximate linear functions. These "zombie units" differ from saturated units in that they facilitate, rather than obstruct, gradient flow and signal propagation. Nevertheless, an abundance of these units inevitably diminishes the model's effective representational capacity."
>
> 5. We have added this sentence to Section 2: "The scores are normalized such that they sum to 1 within a layer. This makes the comparison of neurons in different layers possible."
>
> 6. We have modified the "DrQ network" in Section 4 to "DrQ implementation" in our revision.
>
> 7. We have separated the stationary tasks and non-stationary ones into Figure 2 and 3, and have added the clarification for non-stationarity in the caption of Figure 3. Moreover, two more reward non-stationary tasks, Cheetah-run\_sparsified and Reacher-hard\_sparsified, have also been added (Figure 3), demonstrating consistent patterns of plasticity loss. The sparsification is the same as in Reacher-easy\_sparsified, where the reward is masked to zero during the first half of training.
>
> 8. To strengthen the results in Figure 8-9 (now 10-11), we have added two more reward non-stationary tasks to Appendix C.4/C.5, i.e., Reacher-hard\_sparsified and Cheetah-run\_sparsified. The sparsification is the same as Reacher-easy\_sparsified. The overall trend is similar to that exhibited by Figure 10-11: 1) LayerNorm is able to eliminate dominant neurons and revive a fully saturated layer back to normal; and (2) HL-Gauss can prevent the penultimate layer from being fully saturated, and as long as the reward signal is revealed to the network, the penultimate dormant percentage instantly returns to a low value.

---

> > ### Comment · Reviewer_oc3b · 2026-04-07
> > **Thank you for the changes**
> >
> > Thank you for your hard work. These changes greatly strengthen the paper, and I believe the (more limited) claims are now well-supported.

---

### Review · Reviewer_4DEC · 2026-03-24

**Summary Of Contributions:**

**Summary**

This paper investigates a potential factor leading to capacity loss in deep reinforcement learning (DRL): "dominant neurons". Through theoretical analysis, the authors point out that dominant neurons induce complete saturation in the subsequent layer. Furthermore, the paper establishes a theoretical connection between the degree of neuronal dominance and the degradation of network representational capacity (rank collapse). Empirically, the paper evaluates various mitigation strategies, including Reset, ReDo, Pruning, HL-Gauss, and LayerNorm with Weight Decay (LNWD), on vision-based DM-Control tasks. The results indicate that LNWD achieves better overall performance, while categorical loss (HL-Gauss) performs better in sparse reward environments in sparse reward environments. Finally, the authors provide a continuous-time dynamics analysis, theoretically explaining the mechanism by which categorical loss prevents representation collapse compared to traditional regression loss.

**Strengths**

Novel Perspective: This paper approaches the problem from the angle of "dominant neurons," providing a new analytical dimension to prior research that primarily focused on "dormant neurons".
Combination of Theory and Empiricism: It mathematically links the activation state of single neurons with the rank collapse of network representations. Theorem 3 offers a theoretical explanation for the mechanism by which categorical methods preserve plasticity.
Rigorous Experimental Evaluation: The experimental design is sound, reporting statistical metrics such as IQM and the median, and conducting an ablation study on LNWD.

**Weaknesses**

There is a discrepancy between the assumption of Theorem 1 (bias-free) and the actual network architecture used in experiments (DrQ includes biases).
The threshold set in Definition 1 ($u_i^k>\frac{N_k}{2}$) is relatively high and difficult to satisfy for wider hidden layers.
The experiments are primarily limited to a single algorithm (DrQ) and a fixed CNN+MLP network architecture.
While the paper's motivation revolves around plasticity loss, the empirical evaluation is heavily concentrated on stationary DM-Control tasks, lacking investigation into non-stationary environments or continual learning scenarios.
The paper theoretically derives an inter-layer causality via Theorem 1 (i.e., dominant neurons in layer k cause saturation in layer k+1); however, the empirical section only provides evidence of a macroscopic negative correlation between the degree of dominance and the network's effective rank. This statistical correlation fails to effectively rule out the possibility that the complete saturation of layer k+1 is independently driven by intra-layer competition within that same layer.

**Additional Comments:**

The structure of this paper is clear. Notably, Theorems 1-3 provide a theoretical perspective for analyzing the relationship between categorical loss and representation collapse, enriching the understanding of this problem.

**Audience:**

Yes

**Audience Explanation:**

Plasticity loss is currently one of the key research directions in the field of deep reinforcement learning. Investigating the relationship between "dominant neurons" and network layer saturation holds reference value for understanding reinforcement learning representational dynamics. In addition, the empirical conclusions regarding the robustness of LNWD, as well as the dynamic explanation of how categorical value functions (e.g., HL-Gauss) prevent collapse, offer certain inspirations for engineering practices and theoretical advancements in this field.

**Broader Impact Concerns:**

None. This paper focuses on foundational optimization dynamics and plasticity mechanisms of neural networks in reinforcement learning. It does not introduce novel applications that directly involve ethical, social, or safety risks.

**Claims And Evidence:**

Yes

**Claims Explanation:**

The core claims of the paper are generally supported by theoretical and empirical evidence. Theorems 1, 2, and 3 provide mathematical derivations for the subsequent layer saturation phenomenon, the lower bound of the singular value ratio, and the dynamics of categorical loss, respectively. Empirically, the authors conduct extensive experiments based on the DrQ algorithm across dense and sparse reward DM-Control tasks, with learning curves and scatter plots intuitively demonstrating the negative correlation between the maximum activation score and the effective rank. It should be noted that there is a discrepancy between the bias-free assumption of Theorem 1 and the actual experimental networks containing biases, but this has been addressed in the main text's remarks, making the overall chain of evidence relatively complete.

**Requested Changes:**

1. Clarify the theoretical and practical discrepancy of Theorem 1: The DrQ network in the experiments contains biases. Please supplement an analysis in the main text or appendix discussing what specific impacts the presence of bias terms would have on the derivation of dominant neurons and their consequent saturation of the subsequent layer when the "bias-free" assumption is relaxed.

2. Clarify the statistical significance of Definition 1: Please provide experimental data (e.g., activation score distributions) to explicitly state whether the dominant neurons causing network collapse in sparse reward DM-Control tasks actually satisfy the threshold set by $u_i^k>\frac{N_k}{2}$. If it is not strictly satisfied in practice, please discuss the impact of relaxing this threshold on the generalizability of the theorem.

3. Supplement the perspective of non-stationary environments: Plasticity loss typically manifests more significantly in non-stationary environments or continual learning. It is recommended to supplement relevant analyses or preliminary experiments to observe the evolutionary characteristics of dominant neurons when dynamic shifts occur in the task state distribution or reward function.

4. Expand the scope of evaluation: Currently, the validation of the dominant neuron phenomenon is entirely based on the DrQ (CNN+MLP) architecture. It is recommended to supplement statistical data on at least one different mainstream architecture or algorithm (e.g., SAC without image augmentation, or PPO) to verify the universality of this mechanism.

5. Analyze the negative effects of parameter tuning strategies: Regarding the phenomenon observed in Section 4.1.2 where ReDo and Pruning conversely lead to performance degradation in non-sparse (dense) reward tasks, please supplement corresponding network dynamics hypotheses or mechanistic analyses.

6. Lack of direct empirical evidence for inter-layer causality: Figure 7 in the experiments only reveals a statistical negative correlation between the maximum activation score and the network's effective rank. While this aligns with the inference of Theorem 1, it fails to strictly rule out the possibility that the saturation of layer k+1 is independently triggered by local intra-layer competition among neurons within that same layer. To strengthen the paper's theoretical claims, it is recommended to supplement intervention experiments targeting inter-layer coupling effects. For instance, the authors could consider introducing a single-layer intervention at a specific training stage—artificially constructing a dominant neuron satisfying Definition 1 in a healthy layer—and subsequently tracking the immediate neuronal evolution in layer k+1. Providing such direct observational evidence at the microscopic level would greatly enhance the empirical persuasiveness of Theorem 1 within actual complex networks.

---

> ### Author Response · Authors · 2026-04-06
>
> We thank the reviewer for the constructive feedback. Modifications reflecting the following responses have been made in the revised paper, highlighted in blue.
>
> 1. The zero-bias assumption in layer k+1 can be relaxed to $b_m\leq \mathbb{E}\left[|w_{1m}^{k+1} g^{k}_1(x)|\right] - \sum\_{j \neq 1} \mathbb{E}\left[|w\_{jm}^{k+1}g\_j^{k}(x)|\right]. $
> In other words, the bias of a neuron is no greater than the difference between the dominant neuron and other neurons (multiplied by their corresponding weights). We have added this as a remark to Appendix A.1. Moreover, we observe a phenomenon similar to Figure 2 by manually constructing a dominant neuron in supervised learning (see 6 for the details) where we use a network with non-zero biases.
> 2. Since the width for layer k is 256 in our experiment, most largest activation scores in Figure 2 satisfy the condition in Definition 1. We have added this sentence to Section 3.2.
> 3. As mentioned in Section 4.1.4, Reacher-easy\_sparsified is a reward non-stationary task where the reward is masked to 0 during the first half of training, and is revealed to the agent in the second half. To improve the clarity, we have added the task description in the caption of Figure 2. Moreover, two more reward non-stationary tasks have also been added to Figure 2, demonstrating consistent patterns of plasticity loss.
> 4. We thank the reviewer for the advice. We have added the results of DrQ with ResNet-18 in Appendix C.1 in the revision. As shown by the figure, the phenomenon consistently occurs when the CNN layers are replaced by ResNet-18.
> 5. We agree that this is an interesting phenomenon, as it indicates that intervention costs outweigh the gains in non-sparse reward environments. Although these tasks do exhibit reduced neuron dominance and dormancy, the exact negative impact of ReDo and pruning on learning dynamics is not yet fully understood. We leave the formal characterization of this relationship for future research (added to Conclusion).
> 6. The activation score is partly determined by the data distribution which keeps changing during online RL, and thus it is infeasible to manually construct a dominant neuron during this process. As an alternative, we conduct an intervention experiment on CIFAR10 where the data distribution is fixed. The dominant neuron is created by scaling up the incoming weights of the neuron that has the largest activation score. The results which we have added to Section 3.2 in the revised paper, show that shortly after the dominant neuron is created, its subsequent layer becomes completely saturated, and the network ceases to learn.

---

### Decision · Action_Editor_J4wA · 2026-05-01

**Recommendation:** Accept as is

**Audience:**

Yes

**Audience Explanation:**

Plasticity loss in deep RL is a topic that has garnered a lot of interest in recent years, and there continues to be a substantial amount of work in better understanding and mitigating this issue. This paper is a solid contribution to this line of work and would be of interest to many in the community.

**Claims And Evidence:**

Yes

**Claims Explanation:**

The authors have provided both theoretical and empirical evidence to support their claims. This was strengthened during the rebuttal where the authors added additional synthetic experiments to better support the theoretical results.

All authors are supportive of accepting this work, and I agree with them.